# Inter-comparison between the Aerosol Optical Properties Retrieved by Different Inversion Methods from SKYNET Sky Radiometer Observations over Qionghai and Yucheng in China

Zhe Jiang[1], Minzheng Duan[1*], Huizheng Che[2*], Wenxing Zhang[1], Teruyuki Nakajima[3], Makiko Hashimoto[3], Bin Chen[1], and Akihiro Yamazaki[4]

[1]Institute of Atmospheric Physics, Chinese Academy of Sciences, Beijing 100029, China
[2]State Key Laboratory of Severe Weather (LASW) and Institute of Atmospheric Composition, Chinese Academy of Meteorological Sciences (CAMS), CMA, Beijing 100081, China
[3]Earth Observation Research Center (EORC), Japan Aerospace Exploration Agency (JAXA), Tsukuba, Ibaraki 305-8505, Japan
[4]Japan Meteorological Agency, Meteorological Research Institute, 1-1 Nagamine, Tsukuba, Ibaraki 305-0052, Japan

*Correspondence to*: Huizheng Che(chehz@cma.gov.cn) and Minzheng Duan(dmz@mail.iap.ac.cn)

**Abstract.** This study analyzed the aerosol optical properties derived by SKYRAD.pack versions 5.0 and 4.2 (referred to as V5.0 and V4.2) using the radiometer measurements over Qionghai and Yucheng in China, two new sites of the sky radiometer network (SKYNET). As V5.0 uses an a priori size distribution function (SDF) of a bimodal log-normal function, the volume size distribution retrieved by V5.0 presented bimodal patterns with a 0.1-0.2 μm fine particle mode and a 3.0-6.0 μm coarse particle mode both over Qionghai and Yucheng. The differences of the volume size distributions between the two versions were very large for the coarse mode with a radius of over 5 μm. The single scattering albedos (SSAs) by V5.0 correlated to SSAs by V4.2 with R= 0.88, 0.87, 0.90, 0.88 and 0.92 at wavelengths of 400, 500, 670, 870, and 1020 nm over Qionghai, respectively. The correlation coefficients were around 0.95, 0.95, 0.96, 0.94, and 0.91 at the five channels in Yucheng. An error of ±5% for solid view angle SVA introduced about ±2% differences in retrieved SSA values both by V4.2 and V5.0. An error of ±50% for ground surface albedo Ag caused about 1% averaged differences in retrieved SSA values both by V4.2 and V5.0. With the atmospheric pressure PRS increased by 1%, 2%, 3% and 4%, the averaged differences in SSAs didn't exceed 0.8%.The SSA differences at 500nm between the two versions decreased while AODs increased over both sites. The seasonal variability of the aerosol properties over Qionghai and Yucheng were investigated based on SKYRAD.pack V5.0. The seasonal averaged AOD over Qionghai had higher values in spring, winter and autumn while lower in

summer. The AOD averages were commonly higher in summer and spring than in winter and autumn in Yucheng. The lowest seasonal averaged SSAs were both observed in winter in the two sites. The fraction of the fine aerosol particles was much smaller in summer than the other seasons over Qionghai; the volume distribution of the coarse-mode particle in Yucheng had much large values compared to the fine-mode particle in all seasons.The results can provide validation data in China for SKYNET to continue improving the data-processing and inversion method. The results provide valuable references for continued improvement of the retrieval algorithms of SKYNET and other aerosol observational networks.

**1 Introduction**

Aerosols are well known to have significant impacts on climate change and global hydrologic cycle by absorbing, scattering, and reflecting solar radiation (Hansen et al., 1997; Sun et al., 2017) and participating in cloud processes (Ackerman et al., 2000; Ramanathan et al., 2001; Kaufman et al., 2005; Li et al., 2011; Bi et al., 2014; Zhao et al., 2018a). Aerosols also adversely influence human health and visibility (Samet et al., 2000; Pope lii et al., 2002; Yang et al., 2015; Wang et al., 2017). Aerosol-related problems have drawn a great deal of attention (Cai et al., 2016).

Using a sun/sky radiometer to measure both direct solar beam and angular sky radiance is the most common method for a reliable and continuous estimate of detailed aerosol properties over mega-cities around the world. Several aerosol ground-based observation networks were established to understand the aerosol optical properties, validate the inversion products of satellite remote sensing, and indirectly evaluate their effect on climate (Uchiyama et al., 2005; Takamura and Nakajima, 2004; Nakajima et al., 2007). SKYNET, the focus of this study, is a ground-based research network of using sky radiometers (PREDE Co., Ltd., Tokyo, Japan) with observation sites principally located in Asia and Europe (Che et al., 2014).

The direct solar and angular sky radiance data measured by the sky radiometers are processed to obtain the aerosol optical properties, such as aerosol optical depth (AOD), single scattering albedo (SSA), complex refractive index, and volume size distribution function (SDF) using SKYRAD.pack, which is the official retrieval algorithm of the SKYNET network (Nakajima et al., 1996) having several different versions. SKYNET currently uses the SKYRAD.pack algorithm version 4.2 (Takamura and Nakajima, 2004). The aerosol retrievals derived from SKYRAD.pack version 4.2 algorithm have been used to

investigate the regional and seasonal characteristics of aerosols for climate and environmental studies

and validate satellite remote sensing results (e.g., Kim et al., 2004; Che et al., 2008; Campanelli et al.,

2010; Estellés et al., 2012a; Wang et al., 2014; Che et al., 2018). Recently, a new SKYRAD.pack version

5.0 was proposed to improve SSA retrievals (Hashimoto et al., 2012), there are a few applications of

SKYRAD V5.0 in China, and it was just preliminarily used to retrieve aerosol optical properties over

Beijing in China (Che et al., 2014).

    This study presents the aerosol optical properties over Qionghai and Yucheng by using SKYRAD.pack

V5.0 and V4.2 from SKYNET sky radiometer measurements during February 2013 to December 2015.

This work is designed to achieve the following objectives: (1) investigate the difference of the aerosol

optical properties derived by SKYRAD.pack V5.0 and V4.2 over the two SKYNET sites; and (2) analyze

the seasonal variability of aerosol optical properties over Qionghai and Yucheng based on

SKYRAD.pack V5.0. The results presented in this study provide valuable references for continued

improvement of the retrieval algorithms of SKYNET and other aerosol observational networks.

**2 Site description, instrumentation, and inversion method**

**2.1 Instrumentation**

The sky radiometer (Model POM-02, PREDE Co. Ltd.) was deployed at Qionghai and Yucheng from

February 2013 and February  2013, respectively. The PREDE-POM02 model was equipped with an

InGaAs detector to measure the direct solar irradiance and the sky diffuse radiance at 11 wavelengths,

namely, 315, 340, 380, 400, 500, 675, 870, 940, 1020, 1627, and 2200 nm. The data from five channels at

400, 500, 675, 870, and 1020 nm were used here to retrieve the aerosol optical properties over Qionghai

and Yucheng. The full angle field of view is 1.0°, while the minimum scattering angle of measurement is

approximately 3°. The sky radiance is measured at 24 predefined scattering angles and at regular time

intervals. The sky radiometer operates only during daytime and collects data regardless of the sky

conditions. Its dynamic range is $10^7$. The typical measurement interval of the sky radiance is 10 min. The

Improved Langley (IL) plot method is used in this study to determine the temporal and spectral

calibration constants for direct intensity (F0) with accuracy of about 1.5–2.5 %, depending on the

wavelength (Nakajima et al., 1996; Campanelli et al., 2004). The calibration by IL plot method was

made daily, the variation of F0 due to instrumental drift could be quickly spotted, and then appropriate

corrections to data can be applied exactly from the period in which the deviation occurred (Campanelli et

al., 2004). The calibration method for sky radiance measurements is different from the calibration method for the direct solar irradiance measurements. The solar disk scan method has been routinely used in the SKYNET measurement of the SVA of the sky radiometer by scanning a circumsolar domain (CSD) of ±1° around the sun with every 0.1° interval (Nakajima et al., 1996; Uchiyama et al., 2018).

**2.2 Site description**

The Qionghai site of SKYNET (19.23°N, 110.46°E, 24 m a.s.l.), which was located in the eastern part of Hainan Island, was mainly influenced by East Asia monsoons and typhoons. During summer, the dominant wind is from south to southeast, summer monsoon from the South China Sea and West Pacific brought most of the annual rainfall to the island (Zhu et al., 2005), whereas the winter monsoon from Inner Mongolia carries dry winds to the area (Zhu et al., 2005; Peel et al., 2007; Yin et al., 2002). Annual average rainfall in Qionghai is estimated about 1653.4 mm. Maximum high temperature occurs in July, with monthly average of 28.6 $^o$C, monthly lowest temperature occurs in January, with monthly average of 19.1$^o$C (Yin et al., 2002).

The other measurement site in this study was located in rural Yucheng (36.82°N, 116.57°E, 22 m a.s.l.), Shandong Province, China, which is almost in the centre of the North China Plain. The selected site is in an open field surrounded by farmland. The region belongs to semi-humid and temperate monsoon climate zone, characterized by a mean annual temperature of 21$^o$C and mean annual precipitation of 610 mm mainly distributed in summer months (Chen et al., 2012).Yucheng and the surrounding areas are famous for their agriculture (e.g., wheat and corn) and grazing land (e.g., donkeys and chickens). In addition, the site near 20 to 30 km radius located several factories in the production of inorganic and organic fertilizers (Wen et al., 2015), and the application of fertilisers to farmland emitted a great deal of NH3 (Zhao et al., 2012). Meanwhile, Yucheng is located in the downwind of the Beijing-Tianjin-Hebei region, long-distance transport of sources of industrial pollution and biomass burning contributed significantly to the concentrations of pollutants in Yucheng (Lu et al., 2016). The major chemical compositions in PM2.5 at the two sites were introduced in Section 3.2.1.

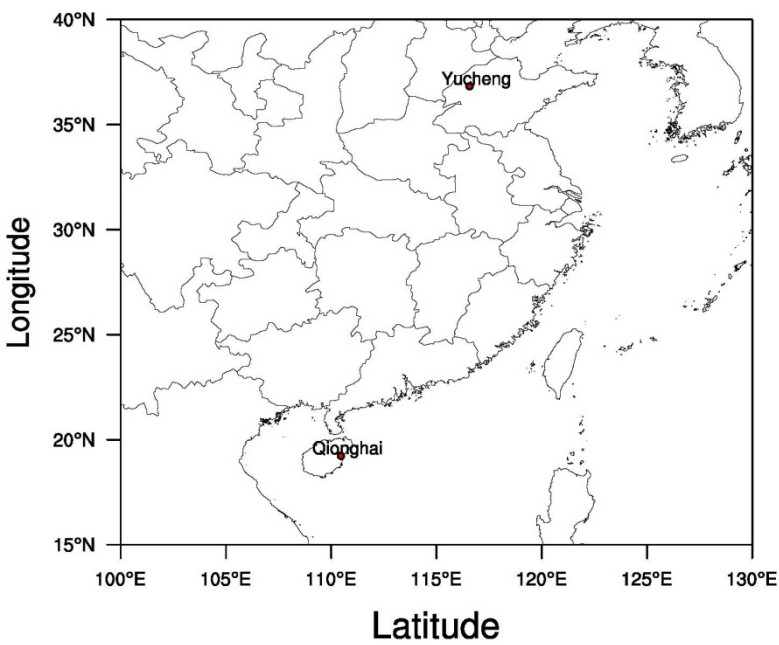

**Figure 1: The locations of the two SKYNET sites in the study**

**2.3 Inversion method**

The aerosol optical properties (i.e., AOD, SSA, complex refractive index, and volume SDF) were
derived in this study by using SKYRAD.pack V4.2 and V5.0. Within the SKYRAD.pack code, the
inversion schemes were used to derive the single scattering term $\beta(\Theta)$ from the measurements of the
normalized sky flux $R(\Theta)$ and retrieve the aerosol SDF $v(r)$ (as a function of particle radius, r) from data
$\beta(\Theta)$ and AOD $\tau$. The inversion of $\beta(\Theta)$ was performed through a non-linear iterative method. Each step
of the loop contained the procedure for retrieving $v(r)$ using a constrained linear or a non-linear iterative
method. Non-sphericity particle model are neither included in V4.2 nor V5.0.

The retrieved $v(r)$ in each iteration step was used as an input parameter for the radiative transfer model
(Nakajima and Tanaka, 1988) to simulate $R(\Theta)$, which was compared with the measured $R(\Theta)$ to
evaluate the root mean square difference $\varepsilon(R)$. The maximum number of iterations and the tolerance
parameter for the convergence of R were set as 20 and 0.1%, respectively.

The retrieval of $v(r)$ from $\beta(\Theta)$ and $\tau$ data in SKYRAD.pack V4.2 was conducted using a constrained
linear method. The inversion method consisted of a linear matrix formulation, in which the solution
stability was controlled by the requirement that it agrees both with the input data and the imposed
weighted constraints (Nakajima et al., 1983).

$$f=Kx+\varepsilon. \quad (1)$$

where f is the vector of the $\beta(\Theta)$ and $\tau$ data, and x is a state vector containing the values of size
distribution $v_i = v(r_i)$ with $r_i$ equidistant on a logarithmic scale (i.e., $\ln(r_{i+1}) - \ln(r_i) =$ const). The
components of vector $\varepsilon$ were the error of each datum, $K = K(m(\lambda))$, a matrix of the kernel coefficients
calculated for the fixed values of the complex refractive index ($m(\lambda)$).

V4.2 used the iterative relaxation method of Nakajima et al. (1983, 1996) to remove the multiple
scattering contribution and derived an optimal solution using a statistical regularization method (Turchin
and Nozik, 1969) by minimizing the following cost function as proposed by Phillips (1962) and Twomey

(1963):

$$e^2=|(f\text{-}Kx)|^2+\gamma|Bx|^2. \quad (2)$$

where B is a smoothing matrix used to generate a priori information that forces the solution x to be a smooth function of ln(r); and $\gamma$ is a Lagrange multiplier coefficient to minimize the first term of the

right-hand side of Eq. (2). The solution of Eq. (1) provided a smooth retrieval of the size distribution v(r) corresponding to the minimum of e2 defined by Eq. (2). In such an approach, both the solution v(r) and $e^2$ depended on the assumed value of the complex refractive index m($\lambda$). The complex refractive index m($\lambda$) in each iteration was also evaluated together with v(r), but the retrieved m($\lambda$) can only be chosen from the predefined set of values.

The m($\lambda$) values in SKYRAD.pack V5.0 were directly included in the state vector x. Eq. (1) becomes non-linear, and V5.0 solved it using the non-linear maximum likelihood method defined by Rodgers (2000). This method was based on the Bayesian theory.

$$p(x|f)=p(f|x)p(x)/p(f) . \quad (3)$$

where p is the probability density function defined as the Gaussian distribution; and x and f denote the

state and measurement vectors, respectively. Accordingly, x was chosen in the maximum likelihood method, such that the posterior probability p(x|f) becomes the maximum under the condition that a priori information is already given. We obtained the following equation in the tangential space to be solved by a Newtonian method by organizing this non-linear equation, such that p(x|f) = max:

$$x_{k+1}=x_k+(U_k^T S_e^{-1} U_k+S_a^{-1})^{-1}[U_k^T S_e^{-1} (f\text{-}f_k)\text{-} S_a^{-1} (x_k\text{-}x_a)] . \quad (4)$$

where $x_k$ is the solution at the $k^{th}$ iteration step; $f_k=f(x_k)$ is an observation modeled using $x_k$ ; $x_a$ is the a priori value of x; $S_e$ is the measurement error covariance matrix; $S_a$ denotes the covariance matrix defined by a priori and state values, $S_a=(x\text{-}x_a)(x\text{-}x_a)^T$; and U is the Jacobi matrix, $\partial f/\partial x$ . The retrieval algorithm used in V5.0 allowed a rigorous retrieval of both the aerosol size distribution and the spectral complex refractive index.

The non-linear inversion has a strong dependence on the estimation of the first-guess solution. Version 5.0 uses an a priori SDF of a bimodal log-normal function as follows:

$$v(r) = \sum_{n=1}^2 C_n \exp \left[-\frac{1}{2}\left(\frac{\ln r - \ln r_{mn}}{\ln S_n}\right)^2\right] . \quad (5)$$

where $r_{m1} = 0.1$ µm; $r_{m2}= 2.0$ µm; $S_1 = 0.4$; $S_2 = 0.8$; $C_1 = 1.0 * 10^{-12}$; and $C_2 = 1.0 * 10^{-12}$, following the reported climate values (Higurashi et al., 2000). For a priori estimates of the refractive index, the

real ($m_r$) and imaginary ($m_i$) parts were set as $m_r = 1.5$ and $m_i = 0.005$, respectively.

SKYRAD V5.0 developed a stricter data quality control method of observation data and cloud screening. The standard process of quality control in SKYNET applies a retrieval error between observations and calculated theoretical values by using retrieval values, $\sigma_{obs}$

$$\sigma_{obs} = \sqrt{W_e \sum_i \left(\frac{\tau_{\lambda_i}}{\tau_{\lambda_i}^{meas}} - 1\right)^2 + W_p \sum_i \sum_j \left[\frac{R_{\lambda_i}(\Theta_j)}{R_{\lambda_i}^{meas}(\Theta_j)} - 1\right]^2} . \quad (6)$$

where ($\tau_{\lambda_i}^{meas}$ and $R_{\lambda_i}^{meas}$) and ( $\tau_{\lambda_i}$ and $R_{\lambda_i}$ ) are measured and retrieved observation vectors for the AOD and relative sky radiance, $N_i$ , $N_j$ , and $N_{total} = N_i+N_i \times N_j$ indicate the number of measured

wavelengths, scattering angles, and their total, respectively, $W_e = W_P = 1/N_{total}$. In V4.2, the data if the value of $\sigma_{obs}$ is larger than 0.2, but $\sigma_{obs}$ is set 0.07 as a threshold for data rejection in V5.0. There are some other differences between V4.2 and V5.0 on the issue of quality control of observation data and cloud screening (Hashimoto et al., 2012).

The cloud screening method in V4.2 relies heavily on the global flux test and needs global irradiance data but, almost uniformly, the observation sites in SKYNET do not conduct an observation of solar irradiance. Furthermore, cirrus contamination data are difficult to remove as cloud-affected data (Hashimoto et al., 2012).

    V5.0 poses a condition regarding the magnitude of the coarse mode of the SDF:

$$C_v \times v(2.4\mu m) < \max\{v(7.7\mu m), v(11.3\mu m), v(16.5\mu m)\}. \quad (7)$$

where $C_v$ is a threshold coefficient to be determined for optimum rejection of cirrus contamination, v(r) is vertically integrated aerosol SDF, as a function of particle radius, r. Based on the analysis of data at the Pune and Beijing sites (Hashimoto et al., 2012), $C_v$ is set as 2 in V5.0 to reject most cirrus contamination cases and pass through dust cases. It is necessary to determine $C_v$ after collecting more 15 cirrus contamination data and dust day data.

**3 Results and discussion**

The results retrieved by SKYRAD.pack V4.2 were used to compare with the results retrieved by SKYRAD.pack V5.0. The inter-comparisons of the volume size distribution, single scatter albedo, and 20 refractive index between V5.0 and V4.2 were based on 1397 measurements for 355 days over Qionghai and 5830 measurements for 473 days over Yucheng. Considering a relatively low retrieval accuracy of SSA when AOD < 0.2 (Dubovik et al., 2000), only the measurements with AOD ≥ 0.2 were selected to be effective values in this study. Figure 2 showed the plots of AOD values at each wavelength derived from the solar direct irradiance between the two versions. High correlation was found with a significant 25 coefficient larger than 0.995 at each band in both sites except 1020nm over Qionghai. High consistency of AODs between V4.2 and V5.0 indicates that the inversion process in V5.0 did not bring about a large change in the retrieved direct solar radiation (Hashimoto et al., 2012). The slight differences between AODs by V5.0 and V4.2 were mainly caused by the very small differences in calibration constant F0. F0 in V4.2 and V5.0 are both determined from sky radiance data by the Improved Langley 30 method. V5.0 adopts more rigorous data processing and cloud detection methods. The sky radiance

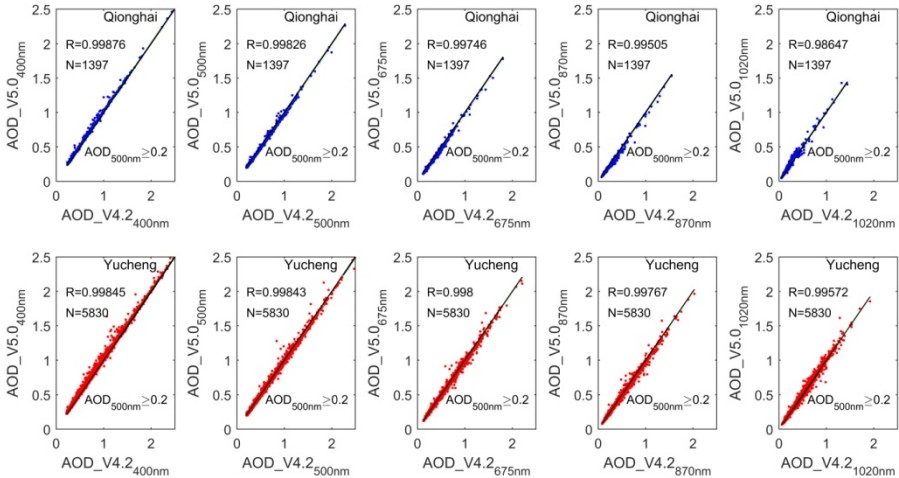

normalized diffuse sky flux data, according to their reliability (Nakajima et al., 1996).

**Figure 2: Scattergrams of the aerosol optical depth (AOD) between SKYRAD V4.2 and V5.0 at wavelengths**

**of 400, 500, 670, 870, and 1020 nm over Qionghai and Yucheng during February 2013 to December 2015.**

**3.1 Inter-comparison of aerosol properties results between SKYRAD V4.2 and V5.0**

**3.1.1 Inter-comparison of the volume size distribution results between SKYRAD V4.2 and V5.0**

Aerosol size properties were one of the most important sources of information for both the observation

and modeling of radiative forcing (Dusek et al., 2006). The volumes at each bin were monthly averaged

during the experiment period, for V4.2 and V5.0 over Qionghai and Yucheng (Figure 3).The SDF by

V4.2 usually showed a predominant peak at the coarse mode with a radius over 10 μm. In Qionghai, the

SDF by V4.2 showed a slightly tri-model pattern in February. There were tri-model patterns with three

peak volumes at radius of 0.026μm, 0.25μm, 16.54μm and 0.25μm, 1.69μm, 11.31μm in volume SDF

by V4.2 in August and September over Yucheng, respectively. As V5.0 uses an a priori SDF of a

bimodal log-normal function (Hashimoto et al., 2012), the volume SDF derived by V5.0 generally

showed the classic bi-mode patterns at both Qionghai and Yucheng. The SDF from V5.0 showed two

peaks at radii of 0.17μm and 5.29μm over both sites. Generally, the SDF retrieved by V4.2 was similar

to V5.0 at radius < 5 μm. The large differences in volume SDF at radius over 5 μm between V4.2 and

V5.0 were mainly related to that the smoothness condition in V4.2 given by Eq. (2) allowed the

retrieved SDF to be distributed beyond 10 μm radius, whereas the strong constraint on the SDF for the

coarse mode particles as shown in Eq. (5) was applied in V5.0 (Hashimoto et al., 2012).

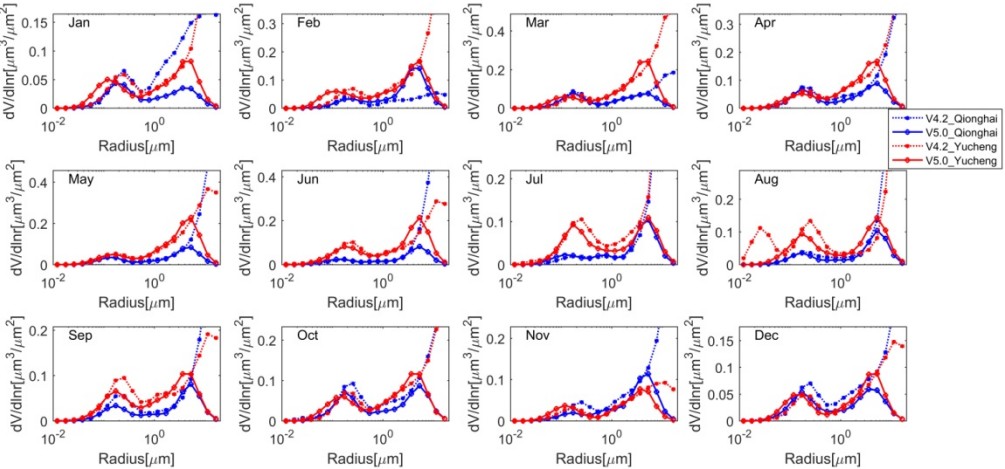

**Figure 3: Retrieved monthly volume size distribution between SKYRAD V4.2 (dotted lines) and V5.0 (solid lines) for Qionghai (blue lines) and Yucheng (red lines) during February 2013 to December 2015**

As shown in Fig.4 and Fig.5, the differences of the retrieved size distribution at smaller size (r <0.05 μm) and larger size (r > 10 μm) were both very large. To avoid unrealistically increasing tails of size distribution appearing due to the very low sensitivity of sky radiometer observations to very small and very large particles, V5.0 introduced this constraint on VOL, the values of the retrieved size distribution of the smallest size classes (r<0.05 μm) and the largest size classes (r > 10 μm) by V5.0 were close to zero.

As shown in Table 1 and Table 2, the percentage difference of the volume size distribution between SKYRAD V5.0 and V4.2 were larger than 50% at smaller size (r <0.025 μm at Qionghai, r <0.017 μm at Yucheng) and larger size (r >10μm at both sites). When the radius is between 0.17-5 μ m, the size distributions retrieved by V5.0 were in good agreement with those by V4.2.

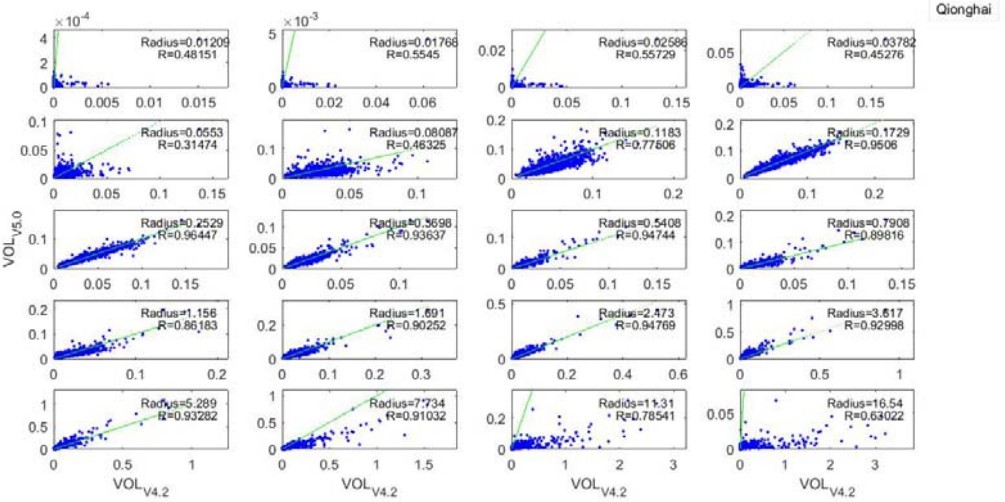

**Figure 4: Scattergrams of volume size distribution between SKYRAD 4.2 and 5.0 in 20 bins over Qionghai during February 2013 to December 2015. Only data with AOD$_{500nm}$>0.2 are shown. The green line means the fitted linear regression curve.**

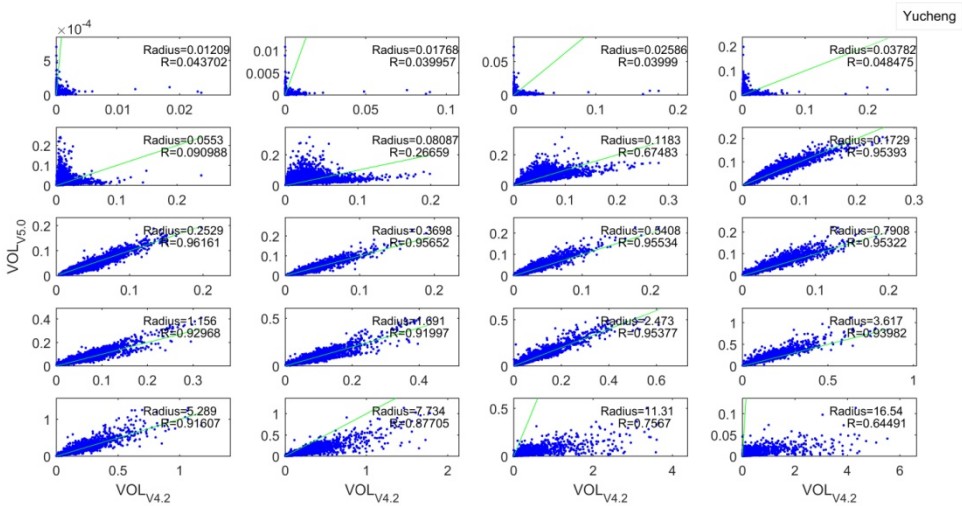

**Figure 5: Scattergrams of volume size distribution between SKYRAD 4.2 and 5.0 in 20 bins over Yucheng during February 2013 to December 2015. Only data with AOD$_{500nm}$>0.2 are shown. The green line means the fitted linear regression curve.**

**3.1.2 Inter-comparison of the single scatter albedo results between SKYRAD V4.2 and V5.0**

10 As a key variable in assessing the climatic effects of aerosols, the SSA is defined as the ratio of the scattering coefficient and the extinction coefficient. It characterized the absorption properties of aerosols and an important quantity in evaluating aerosol radiative forcing. The SSA value is mostly dependent on the shape, size distribution, and concentration of the aerosol particles.

Tables 3 and 4 presented average single scattering albedo and refractive index for SKYRAD 5.0 and

4.2 and the differences between the two versions over Qionghai and Yucheng during February 2013 to December 2015, respectively. The differences of SSA between SKYRAD V5.0 and V4.2 at 400, 500, 675, 870, and 1020 nm over Qionghai were -0.0009(−0.1057%), -0.0028(-0.2984%), -0.0072(−0.7596%), -0.0077(−0.809%), and 0.0039(0.4443%), respectively. The standard deviations of absolute differences were 0.0268, 0.0287, 0.0283, 0.0332 and 0.0454, respectively. Over the Yucheng station, the SSA retrieved from V5.0 were approximately 0.0142(1.5646%), 0.0008(0.0873%), 0.0064(0.6766%), 0.0101(1.1048%) lower than those from V4.2 at 400, 675, 870, and 1020 nm, respectively, but 0.0059(0.6408%) higher than those from V4.2 at 500 nm. The standard deviations of absolute differences at 400, 500, 675, 870, and 1020 nm were 0.0188, 0.018, 0.0208, 0.0267 and 0.0421, respectively.

Figure 6 presented the compared results of SSA between SKYRAD V4.2 and V5.0 at wavelengths of 400, 500, 670, 870, and 1020 nm over Qionghai and Yucheng during February 2013 to December 2015. As shown in Fig. 6, SSAs by V5.0 correlated to SSAs by V4.2 with R= 0.88, 0.87, 0.90, 0.88 and 0.92 at wavelengths of 400, 500, 670, 870, and 1020 nm over Qionghai, respectively. Although the correlation coefficient was highest at 1020 nm in Qionghai, their patterns were more scattered. The SSA values computed from V5.0 had correlation coefficients around 0.95, 0.95, 0.96, 0.94, 0.91 with those from V4.2 at wavelengths of 400, 500, 670, 870, and 1020 nm over Yucheng. Based on the comparison results over the two sites, in most cases V5.0 retrieved lower SSA values SSAs than V4.2. V5.0 tends to underestimate the SSA due to underestimation of the coarse aerosols when the a priori SDF for constraint tends to be close to zero for radii larger than 10μm. Hashimoto et al. (2012) found that the lack of a large coarse part in the SDF causes overestimation of sky radiance at all observation angles. It is likely that V5.0 works to decrease the SSA value to dim the sky radiance in the calculation when a tight constraint on the SDF for particles with radius over 10 μm is applied.

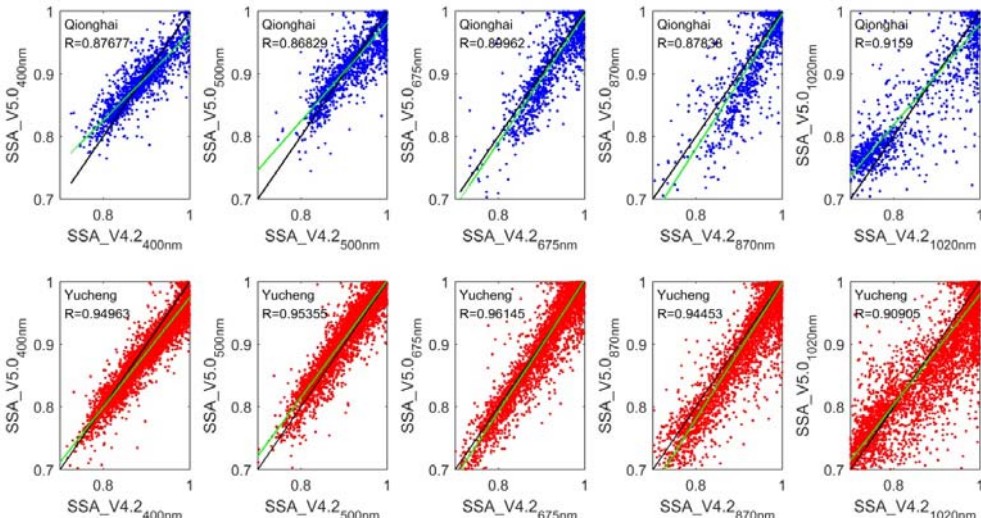

**Figure 6: Scattergrams of the single scattering albedo between SKYRAD 4.2 and 5.0 at wavelengths of 400, 500, 670, 870, and 1020 nm over Qionghai and Yucheng during February 2013 to December 2015. Only data with $AOD_{500nm}>0.2$ are shown. The green line means the fitted linear regression curve.**

**3.1.3 Inter-comparison of the refractive index results between SKYRAD V4.2 and V5.0**

The averaged $m_i$ retrieved from V5.0 at all wavelengths were systemically higher than those by V4.2 over Qionghai (Table 3). The mean values of $m_i$ retrieved from V4.2 were approximately 0.0019, 0.0009, 0.0015, 0.0016, and 0.0021 lower than those from V5.0 for the five channels of 400, 500, 675, 870, and 1020 nm, respectively, over Qionghai. The standard deviations of absolute differences were 0.0038, 0.0039, 0.0038, 0.0041 and 0.0054, respectively. The averaged $m_i$ retrieved by V5.0 was 0.0015, 0.0007, 0.0013 and 0.0017 higher at 400, 675, 870, and 1020 nm wavelengths, respectively, but 0.0005 lower at 500nm, than those retrieved by V4.2 in Yucheng. The standard deviations of absolute differences at 400, 500, 675, 870, and 1020 nm were 0.003, 0.0029, 0.003, 0.0035 and 0.0048, respectively. As shown in Fig.7, the $m_i$ values by V5.0 were linearly correlated with $m_i$ by V4.2 with R=0.8947, 0.8661, 0.8658, 0.8370, 0.9131 at wavelengths of 400, 500, 675, 870 and 1020 nm in Qionghai. The correlation coefficients between $m_i$ by V5.0 and those by V4.2 at the five wavelengths were all higher than 0.89 over Yucheng.

Complex refractive index in V4.2 can only be chosen from the predefined set of values. $m_r$ in V5.0 were directly included in the state vector , including constraints on the complex refractive index. As a priori estimation, $m_r$ usually be set as 1.5. As shown in Fig. 8, $m_r$ have almost random differences over two sites, the difference in $m_r$ between the two versions was greater than that in $m_i$ (as shown in

Tables 3 and 4). The mean $m_r$ from V5.0 were approximately 0.0359 (2.5363%), 0.0118 (0.8263%), and 0.0477 (3.2598%) higher at 400, 500, and 1020 nm, but 0.0019 (0.1321%), 0.0116(0.7926%) lower at 675 and 870 nm than those from V4.2 over the Qionghai station. The results for $m_r$ showed that $m_r$ at wavelengths of 400, 500, 675 and 870 nm by V5.0 were lower than those by V4.2, but larger than those

5  by V4.2 at 1020 nm over Yucheng. The averaged percentage differences of the mean $m_r$ obtained using the two versions were all within 3.26% both at Yucheng and Qionghai. the correlation coefficients between $m_r$ by V5.0 and those by V4.2 at the five wavelengths were all poorer than 0.63 at the two sites.

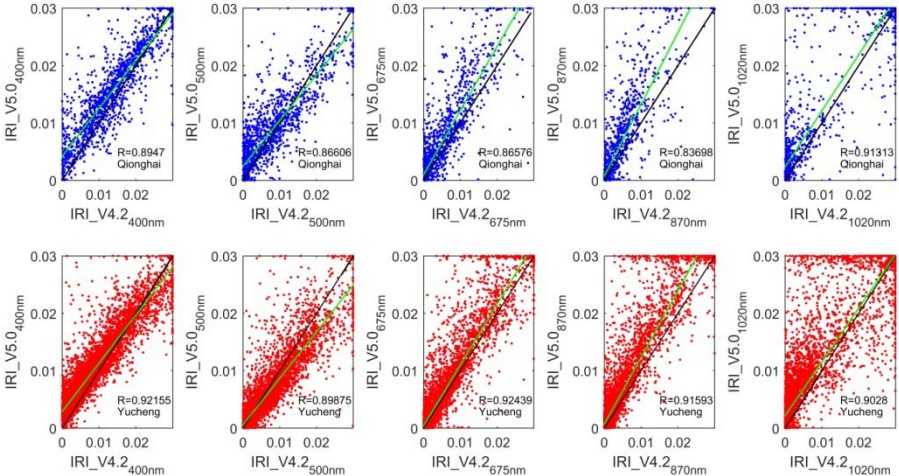

10  **Figure 7: Scattergrams of the imaginary part of the complex refractive index ($m_i$) results between SKYRAD 4.2 and 5.0 at wavelengths of 400, 500, 670, 870, and 1020 nm over Qionghai and Yucheng during February 2013 to December 2015. The green line means the fitted linear regression curve.**

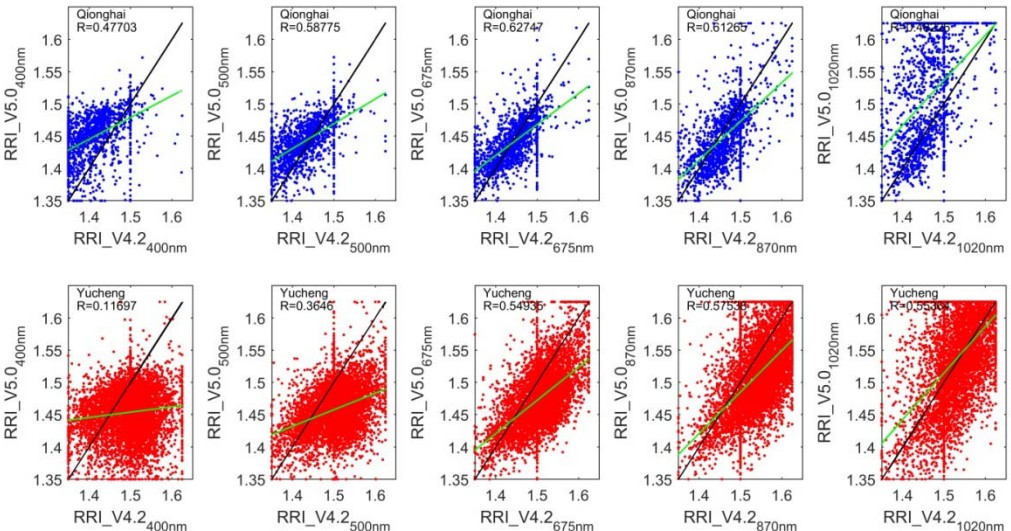

**Figure 8: Scattergrams of the real part of the complex refractive index ($m_r$) results between SKYRAD 4.2 and 5.0 at wavelengths of 400, 500, 670, 870, and 1020 nm over Qionghai and Yucheng during February 2013 to December 2015. The green line means the fitted linear regression curve.**

**3.2 Sensitivity tests**

**3.2.1 Sensitivity tests for the main causes of error in the SSA and AOD retrieval by V5.0 and V4.2**

The accurate retrieval of the SSA is more difficult than estimation of the value of aerosol optical thickness (AOT) and size distribution (Loeb and Su, 2010; McComiskey et al., 2008). The errors associated with Ag (ground surface albedo) and SVA (solid view angle), and the amount of aerosols in the atmosphere, are the causes of error in the SSA, and these errors should cause both underestimation and overestimation of SSA (Hashimoto et al., 2012). The current SKYRAD package method underestimates the SVA by 0.5% to 1.9 % (Uchiyama et al., 2018). The accuracy of calibration constant (F0) determines the inversion accuracy of the amount of aerosols.

Based on the measurements over the two sites in January, April, July and October, 2014, several sensitivity tests were carried out to test the magnitude of the change in SSA. We assumed an error of ±5% for calibration constant $F_0$, ±5% for solid view angle SVA, ±50% (±0.05) for ground surface albedo $A_g$. We compared the differences in retrieved SSA values at a wavelength of 0.5 μm between cases with and without the assumed errors, defined as {[SSA (with assumed error)-SSA (no assumed error)]/[SSA (no assumed error)]}.

As shown in Fig. S1, when we assumed an error of +5% for calibration constant $F_0$, the averaged differences in SSAs $_{V4.2}$ and SSAs $_{V5.0}$ over Qionghai were -2.82% and -3.42 %, while the $F_0$ reduced by 5%, the averaged differences in SSAs $_{V4.2}$ and SSAs $_{V5.0}$ were +4.46% and+3.85 % at Qionghai site. In Yucheng, an error of +5% in $F_0$ causes -2.76% and -3.00% averaged error in SSAs $_{V4.2}$ and SSAs $_{V5.0}$, respectively, when we introduced a -5% error in $F_0$, there was about 2.76% and 3.08% averaged difference in retrieved SSAs by V4.2 and V5.0.

The SVA is related to the sky radiance, and errors in the SVA will affect the SSA results. Figure S2 shows that an error of ±5% for solid view angle SVA introduced about ±2% differences in retrieved SSA values over both two sites. The averaged differences in SSAs $_{V4.2}$ and SSAs $_{V5.0}$ in Qionghai were around 0.6% and 0.36 % when we assumed errors of ±5% for SVA. The sensitivity tests were based on 60 measurements in Qionghai and 607 measurements in Yucheng. The averaged differences in SSAs $_{V4.2}$ and SSAs $_{V5.0}$ over Yucheng were about 0.4% and 0.2 %. The differences in SSAs $_{V5.0}$ were

lower than those in SSAs $_{V4.2}$ both in Qionghai and Yucheng.

Although the value of $A_g$ depends on wavelength and ground conditions, the $A_g$ values used in data processing in V4.2 and V5.0 were both set to 0.1for each wavelength. As shown in Fig.S3, When $A_g$ was increased by 0.05 compared to the initial value 0.1, the SSAs $_{V4.2}$ and SSAs $_{V5.0}$ in Qionghai were about 0.88% and 1.14% smaller than the retrieved SSA results without the assumed $A_g$ errors. However, when the $A_g$ reduced by 0.05, the SSAs $_{V4.2}$ and SSAs $_{V5.0}$ in Qionghai are about 0.98% and 1.16% larger than those from the results without the assumed $A_g$ errors. While over Yucheng site, an error of +5% in $A_g$ causes -0.97% and -0.98% averaged error in SSAs $_{V4.2}$ and SSAs $_{V5.0}$, respectively, when we introduced -5% in $A_g$, there was about +0.93% and +0.95% averaged difference in retrieved SSAs by V4.2 and V5.0.

For the AOD at the wavelength of 0.5 μm, there were no differences when we introduced an error in $A_g$ and SVA, but there were noticeable differences when we introduced a ±5% error in $F_0$ as shown in Fig.S4. The averaged errors in retrieved AODs $_{V4.2}$ and AODs $_{V5.0}$ that caused by +5% error in $F_0$ were +4.00% and +4.83% in Qionghai, +3.25% and +4.13% in Yucheng. The averaged errors in retrieved AODs $_{V4.2}$ and AODs $_{V5.0}$ due to -5% errors in $F_0$ were -4.38% and -4.73% in Qionghai, -3.32% and -4.20% in Yucheng. The differences in AODs $_{V5.0}$ were larger than those in AODs $_{V4.2}$ over the two sites.

We also investigated the differences in SSAs due to the assumed errors in atmospheric pressure (PRS). PRS was considered as 1.00 (atm) as the experimental group, while values of PRS sequentially changed by 1% were regarded as the control group. As shown in Fig.S5, the averaged differences of SSA in four control groups by V5.0 were all smaller than those by V4.2 over both sites. With the PRS increased by 1%, 2%, 3% and 4%, the averaged absolute differences of SSAs over Yucheng by V5.0 were 0.17%, 0.21%, 0.23%, and 0.22%; the averaged absolute differences of SSAs over Yucheng by V4.2 were 0.44%, 0.46%, 0.47%, and 0.50%. In Qionghai, the averaged absolute differences of SSAs by V5.0 were 0.18%, 0.21%, 0.17% and 0.26%; the averaged absolute differences of SSAs by V4.2 were 0.66%, 0.56%, 0.58%, and 0.78%. The differences in SSAs $_{V4.2}$ were larger than those in SSAs $_{V5.0}$ over the two sites.

On the basis of the above sensitivity tests, it is concluded that an error in the calibration constant ($F_0$) causes an error in both retrieved SSA and AOD. However, according to a reported comparison of calibration constants from SKYNET with those from AERONET, the improved Langley method

adopted by SKYNET seems to yield accurate calibration constants (Campanelli et al., 2004; Hashimoto et al., 2012). An error of ±5% for solid view angle SVA introduced about ±2% differences in retrieved SSA values both by V4.2 and V5.0, and the differences in $SSAs_{V5.0}$ were lower than those in $SSAs_{V4.2}$ over both two sites. The sensitivity tests results indicate that overestimation or underestimation in the

Ag results in underestimation or overestimation of the SSA, respectively. An error of ±50% for ground surface albedo Ag caused about 1% averaged differences in retrieved SSA values both by V4.2 and V5.0. For the AOD at the wavelength of 0.5 μm, there were no differences when we introduced an error in $A_g$ and SVA, but there were noticeable differences when we introduced a ±5% error in $F_0$. The averaged differences in retrieved SSA values due to ±5% error in $F_0$ varied from 3% to 5%. With

the atmospheric pressure PRS increased by 1%, 2%, 3% and 4%, the averaged differences in SSAs didn't exceed 0.8%, and the differences in $SSAs_{V4.2}$ were larger than those in $SSAs_{V5.0}$ over the two sites.

**3.2.2 Sensitivity tests for the parameters linked to the SSA differences between the V5.0 and V4.2**

The most different physical process between V4.2 and V5 is a derivation of particle size

distribution. When a large amount of coarse particles of the dust-like aerosol type with radius greater than 10 μm existed , the numerical tests performed by Hashimoto et al showed that V4.2 could retrieve the SDF relatively well, including the coarse mode, in comparison with V5.0, because the smoothness condition given by Eq. (2) allowed the retrieved SDF to be distributed beyond 10 μm radius, on the other hand, V5.0 underestimated the coarse mode of the SDF because of the strong SDF constraint

condition given by Eq. (5) with a small model radius $r_{m2} = 2.0$ μm for the coarse mode SDF (Hashimoto et al., 2012). So we have compared the differences between retrieved SSAs at 500 nm byV5.0 and V4.2 when set $r_{m2} = 1.5, 1.8, 2.0$(default), 2.5 and 3.0 in Skyrad.pack V5.0 based on the measurements in 2014. As shown in Fig.9, SSAs by V5.0 correlated to SSAs by V4.2 with R= 0.860, 0.837, 0.855, 0.809 and 0.826 when $r_{m2} = 2.0$(default), 1.5, 1.8, 2.5 and 3.0 in V5.0 over Qionghai, respectively. The SSA

values computed from V5.0 had correlation coefficients around 0.940, 0.928, 0.928, 0.921, 0.924 with those from V4.2 when $r_{m2} = 2.0$(default), 1.5, 1.8, 2.5 and 3.0 in V5.0 in Yucheng. The correlation coefficient between SSA by V5.0 and V4.2 was the highest while setting $r_{m2}$ as 2.0 (the default value) in V5.0 at the two sites.

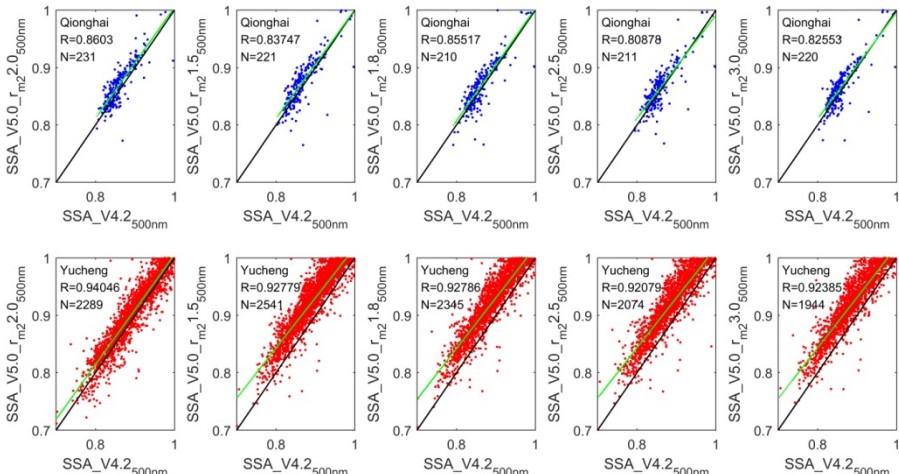

**Figure 9: Scattergrams of retrieved SSA between SKYRAD V4.2 and V5.0 when $r_{m2}$ =2.0(default), 1.5, 1.8, 2.5 and 3.0 for Qionghai (a) and Yucheng (b) in 2014.$r_{m2}$ represents the model radius for the coarse mode SDF.**

We also investigated whether the total amount of aerosols in the atmosphere were linked to the differences in SSA between the two versions. As shown in Fig.10, the SSA differences at 500nm between the two versions (defined as: SSA_V5.0$_{500nm}$ - SSA_V4.2$_{500nm}$) decreased while the corresponding AODs at wavelengths of 500 nm by V5.0 increased at the two sites. When the AOD was high (in this study the threshold was set to 0.5 for AOD$_{500nm}$), SSA retrieved by V5.0 had a good comparison with those by V4.2. It is well known that the inversion products have a very high uncertainty in cases of very low aerosol burdens; the retrieval error in SSA rapidly increases with decreasing AOD, especially in parameters such as the imaginary part of the refractive index (Dubovik et al., 2000).

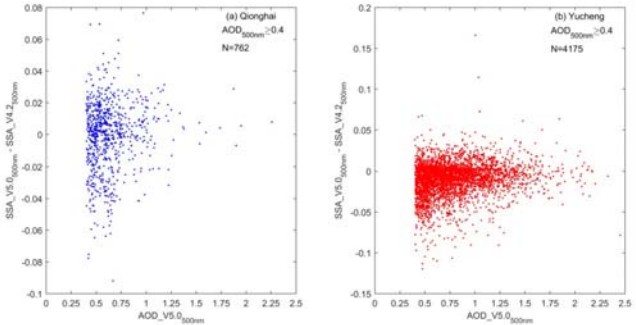

**Figure 10: Scattergrams of the SSA differences at 500nm between V5.0 and V4.2 (defined as: SSA_V5.0-SSA_V4.2) and the corresponding AODs at wavelengths of 500 nm by V5.0 during February 2013 to December 2015.**

Base on the inter-comparison results in Section 3.1 and the sensitivity tests in Section3.2, we couldn't get the conclusion that V5.0 is definitely better than V4.2. We haven't yet got other measurements in the two sites to help us prove that V5.0 is better than V4.2. The most different physical process between V4.2 and V5 is a derivation of particle size distribution. On the one hand, V5.0 tends to be robust to the cloud contamination, owing to inversion constraint by a priori SDF which filters out coarse particles to simulate cloud-scattered radiation. Some tests by Hashimoto et al (2012) showed that the SDF setting in V5.0 was useful for detecting ill-conditioned data caused by cirrus contaminations, horizontally and/or temporally inhomogeneous aerosol stratification, and so on (Hashimoto et al., 2012). On the other, due to a priori SDF for constraint tends to be zero for radii larger than 10µm, V5.0 will underestimate the coarse mode aerosols when a large amount of coarse particles of the dust-like aerosol type with radius greater than 10 µm exits. Estellés et al. (2018) found underestimation of the coarse aerosols by the V5.0 in African dust storm cases, whereas V4.2 retrieved coarse mode SDF similar to the observed one (Estellés et al., 2018).

Considering that V5.0 adopts more rigorous data processing and cloud detection methods, and the SSA and $m_i$ had high correlation coefficients between V4.2 and V5.0 with default the coarse mode radius $r_{m2}$ value in V5.0 based on the above comparison results, we chose the retrieved results by V5.0 to analyze the seasonal variability of the aerosol optical properties over Qionghai and Yucheng in the following section.

### 3.3 Seasonal variability of the aerosol optical properties over Qionghai and Yucheng based on SKYRAD.pack V5.0

The analysis of the 500 nm channel was chosen because it was widely quoted in sun photometric and remote sensing applications and generally representative of visible band wavelengths (Estellés et al., 2012b). Four seasons were considered in this paper (i.e., spring (March-May), summer (June-August), autumn (September-November), and winter (December-February)) to investigate the seasonal variations of the aerosol optical properties over Qionghai and Yucheng based on SKYRAD.pack V5.0.

#### 3.3.1 AOD

The AOD is representative of the aerosol loading in the atmospheric column and important for the identification of the aerosol source regions and the aerosol evolution.

The AOD showed a distinct seasonal variation over both Qionghai and Yucheng. Figure 12a showed

that the seasonal averaged AOD over Qionghai had higher values in spring, winter and autumn while lower in summer. During summer, the dominant wind is from South to Southeast (Zhu et al., 2005), the main emission source was from the South China Sea and West Pacific, in addition, seasonal upwelling off the east coast of Hainan Island was strongest in summer (Li et al., 2018) which was conducive to

pollutant diffusion, meanwhile rich precipitation in summer was effective for eliminating aerosols, so seasonal averaged value of AOD in summer was the lowest in Qionghai, whereas AOD in spring was higher than other seasons. Southerly and northeasterly winds are both prevailing in spring over Qionghai (Liu et al., 2018), so long distance transport and emissions from surrounding areas were probably both the main pollutant sources.

The maximum AOD average of 0.99 occurring in summer, several factories which produced inorganic and organic fertilizers located, the stronger sunlight in summer accelerated the photochemical reaction and enhanced the formation of fine particulate nitrate (Wen et al., 2015), the humidity in summer is higher than other seasons over Yucheng (Meng et al., 2007), high humidity combined with large fractions of hygroscopic chemical components (e.g. sulfate, nitrate, ammonium, and some organic

matters) can enhance light extinction and haze intensity the scattering coefficient of secondary inorganic aerosols (such as sulfate, nitrate and ammonium) (Tao et al., 2017). The prevailing winds in Yucheng were from the northwest in winter and spring, and Yucheng was in the downwind of Hebei province where located many industrial enterprises emitted pollutants including secondary inorganic aerosols (Tao et al., 2017; Zhao et al., 2018c).AOD was higher in spring than in autumn and winter

likely related to the long-range transportation of dust from northern/northwestern China and pollutants emitted from enterprises in Hebei (Tan et al., 2012; Tao et al., 2017).

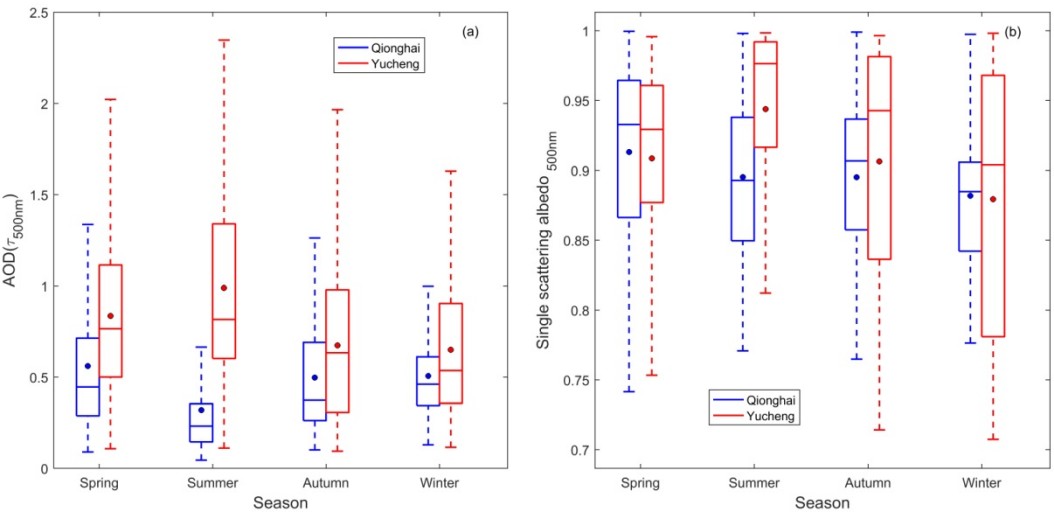

**Figure 12: Seasonal variations in the AOD (a) and the single scattering albedo (SSA) (b) based on SKYRAD V5.0 over Qionghai and Yucheng for the period from February 2013 to December 2015. The boxes represent the 25th to 75th percentiles of the distributions while the dots and solid lines within each box represent the means and medians, respectively.**

### 3.3.2 SSA

Figure 12b shows the seasonal averaged SSA at 500 nm for Qionghai and Yucheng during February 2013 to December 2015. The seasonal averaged SSA values were approximately 0.91, 0.90, 0.90, and 0.89 in spring, summer, autumn, and winter, respectively. The lowest seasonal average SSA was observed in winter, which was probably attributable to the regional transport of the air masses originated from the regions outside of Hainan province in Eastern China, where a great amount of coal was used for industrial enterprises and emitted high concentrations of OC and EC (Liu et al., 2018). In Yucheng, the seasonal pattern of SSA was consistent with AOD, the lowest seasonal average SSAs were also observed in winter due to carbonaceous aerosols increasing by heating activities and biomass burning, seasonal average contributions of carbonaceous aerosols were evidently higher in cold seasons than in warm seasons (Tao et al., 2017). High concentrations of fine particulate nitrate were frequently observed in summer in Yucheng (Wen et al., 2015), likely to cause the high SSA in summer.

### 3.3.3 Volume Size Distribution

Figures 13a and b show the seasonal averaged volumes of the different aerosol particle size distributions ($dv$/dln$r$) in Qionghai and Yucheng. The aerosol volume size distributions were typical bimodal patterns during each season at the Qionghai and Yucheng sites. Figures 11a and 11b show that there was a larger contribution of coarse-mode particles to the aerosol volume compared with the fine-mode particles at the two sites. The fine mode showed a peak at a radius of 0.17 μm in all seasons over Qionghai. The coarse mode was characterized by a peak at a radius of 5.29 μm in spring, summer, and autumn and 3.62 μm in winter. As shown in Fig. 11a, the fraction of the fine aerosol particles was much smaller in summer than other seasons, the summer meteorological conditions such as high wind speeds, high mixing heights, and the fresh air masses originated from or passed through the sea, which may be contributable to the decrease of anthropogenic pollutant concentrations (Liu et al., 2018) and introducing some sea salt particles of a relatively large size. The seasonal averaged peak of fine mode and coarse mode SDF were both higher in winter than other seasons as shown in Fig.11a. The stable atmospheric circulation provides a stable atmospheric environment background, which is not conducive

to the diffusion of pollutants; moreover, continuous low-level northeast wind facilitates the

transportation of pollutants from the inland to Hainan in winter (Wu et al., 2011; Liu et al., 2018).

As shown in Fig. 13b, the coarse-mode particle in Yucheng had a relatively large value compared to

the volume distribution of the fine-mode particle. The aerosol was not only from winter heating but

also from regional transport in winter (Tao et al., 2017; Zhao et al., 2018c). The volume of the coarse

aerosol particles relative to the whole was much larger in spring than other seasons in Yucheng

probably because of the presence of the dust particles transported from the northwest of China and

pollutants emitted from enterprises in Hebei (Tao et al., 2017).

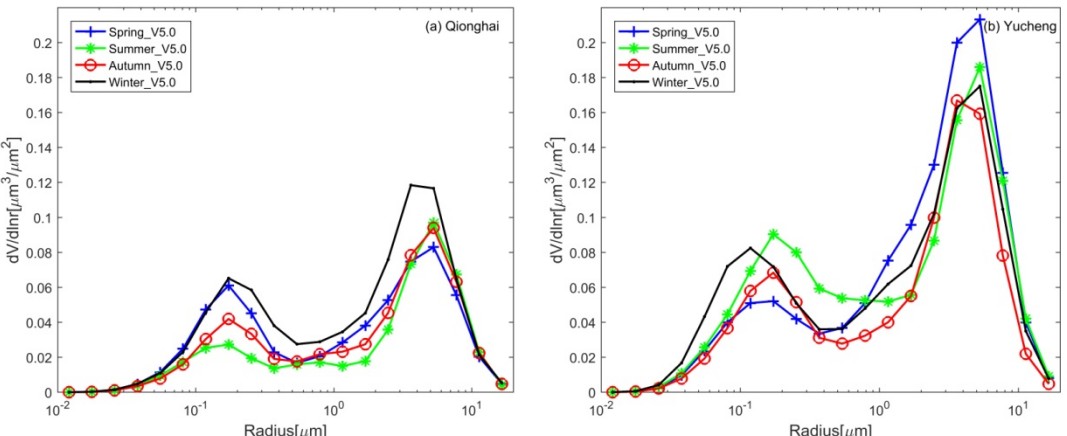

**Figure 13: Seasonally averaged volumes of the different aerosol particle size distributions based on SKYRAD**

**V5.0 over Qionghai (a) and Yucheng (b) for the period from February 2013 to December 2015.**

**3.3.4 Refractive index**

The real part of the refractive index ($m_r$) represents scattering. A higher $m_r$ indicates a higher scattering.

The imaginary part of the refractive index ($m_i$) represents absorption of the aerosols and is an important

quantity in evaluating the aerosol radiative forcing.

Figure 14a showed the seasonal variation of the real part of the refractive index ($m_r$) at 500 nm over

Qionghai and Yucheng. The seasonal averages of $m_r$ at 500 nm were 1.45, 1.46, 1.45, and 1.43 in

spring, summer, autumn, and winter in Qionghai, respectively. The $m_r$ showed a maximum of

approximately 1.47 in spring and a minimum of approximately 1.45 in summer in Yucheng. Figure 14b

presented the seasonal variation of the imaginary part of the refractive index at 500 nm over Qionghai

and Yucheng. The results of imaginary part of complex refractive index ($m_i$) were both the highest in

winter over the two sites. Aerosol absorption coefficient was mainly determined by elemental carbon

(EC) mass concentration and its coating (Tao et al., 2017), heating activities and biomass burning

induced higher carbonaceous aerosols in winter in Yucheng.

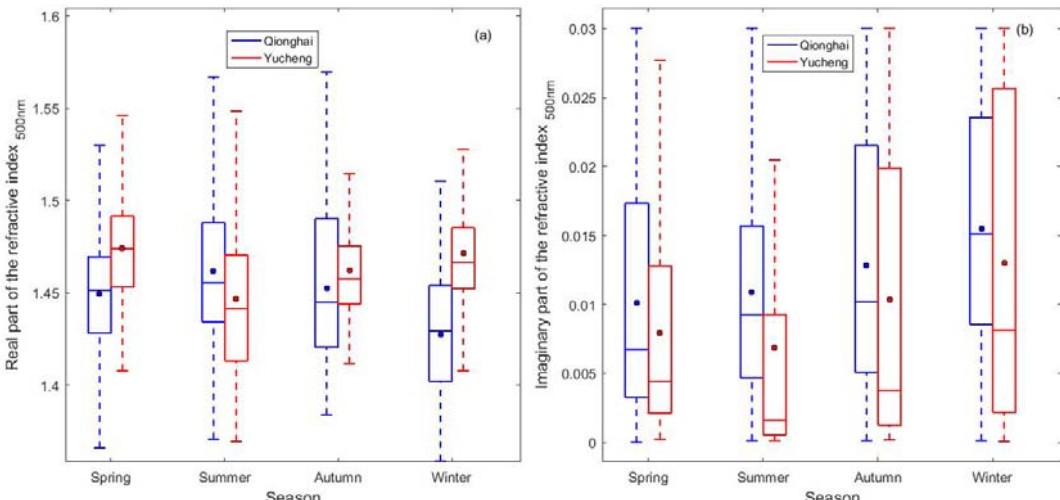

**Figure 14: Seasonal variations in the real part of the refractive index (a) and the imaginary part of the**

**refractive index (b) based on SKYRAD V5.0 over Qionghai and Yucheng for the period from February 2013**

**to December 2015. The boxes represent the 25th to 75th percentiles of the distributions while the dots and**

**solid lines within each box represent the means and medians, respectively.**

**4 Summary**

The aerosol optical properties over the two new SKYNET sites of Qionghai and Yucheng in China were continuously investigated over two years using the RREDE-POM02 sky radiometer measurements. As V5.0 used an a priori SDF of a bimodal log-normal function, the volume size distribution retrieved by V5.0 presented an overall bimodal pattern with a 0.10-0.20 μm fine particle mode and a 3.0–6.0 μm coarse particle mode both over Qionghai and Yucheng. The correlation coefficients between SSAs by V5.0 and by V4.2 were around 0.88, 0.87, 0.90, 0.88 and 0.92 at wavelengths of 400, 500, 670, 870, and 1020 nm over Qionghai, respectively. The SSA values computed from V5.0 had relatively high correlation coefficients with R= 0.95, 0.95, 0.96, 0.94, 0.91 at wavelengths of 400, 500, 670, 870, and 1020 nm in Yucheng. The correlation coefficients between $m_i$ by V5.0 and those by V4.2 at the five wavelengths were all higher than 0.89 over Yucheng.

On the basis of the sensitivity tests, it is concluded that an error in the calibration constant (F0) causes an error in both retrieved SSA and AOD. The averaged differences in retrieved SSA values due to ±5% error in F0 varied from 3% to 5%. An error of ±5% for solid view angle SVA introduced about ±2% differences in retrieved SSA values both by V4.2 and V5.0. Overestimation

or underestimation in the Ag results in underestimation or overestimation of the SSA. An error of ±50% for ground surface albedo Ag caused about 1% averaged differences in retrieved SSA values both by V4.2 and V5.0. With the atmospheric pressure PRS increased by 1%, 2%, 3% and 4%, the averaged differences in SSAs didn't exceed 0.8%.

Sensitivity tests showed that the correlation coefficient between SSAs at 500nm by V5.0 and V4.2 was higher while setting $r_{m2}$ as 2.0μm (the default value) in V5.0 than $r_{m2}$ = 1.5, 1.8, 2.5 and 3.0 μm at the two sites. The SSA differences at 500nm between the two versions decreased with the increase of the corresponding AODs at the two sites.

Based on SKYRAD.pack V5.0, the seasonal variations of the aerosol optical properties over

Qionghai and Yucheng were investigated. The seasonal patterns of AOD were quite different between the two stations. The AOD showed high values in spring, autumn and winter but decreased to minimum in summer over Qionghai, likely related to summer monsoon from the South China Sea and West Pacific that brought most of the annual rainfall to the island, whereas the winter monsoon from Inner Mongolia carried the air masses from the mainland China to Qionghai. In Yucheng, the maximum

seasonal averaged AOD and SSA both appeared in summer due to the hygroscopic effects. The fraction of the fine aerosol particles over Qionghai was much smaller in summer probably related to wet deposition, more precipitation in the summer can lead to more efficient removal of aerosol. The volume of the coarse aerosol particles relative to the whole in spring was much larger than other seasons in Yucheng, probably due to the presence of the dust particles transported from the northwest of China

and pollutants emitted from enterprises in Hebei. The location and distribution of major industrial sources, intensity of local minor sources such as winter heating, and prevailing wind directions together caused the different magnitudes of seasonal variations among the two sites discussed above.

The comparison results between the aerosol optical properties retrieved by SKYRAD5.0 and SKYRAD4.2 were very different over the two SKYNET sites. The results can provide validation data

in China for SKYNET to continue improving data-processing and inversion method. Meanwhile, the results can promote the integration of more Chinese observation stations into international network.

**Data availability.** The sky radiometer data at Qionghai and Yucheng, China are available on request by contacting the first author of the paper (jiangzhe@mail.iap.ac.cn).

**Author contributions.**   Zhe Jiang and Huizheng Che designed the present study, Minzheng Duan and Wenxing Zhang performed observation, Zhe Jiang analyzed data and wrote the paper, with support from all the authors. Teruyuki Nakajima designed the inversion method and Makiko Hashimoto improved the inversion method. Teruyuki Nakajima, Makiko Hashimoto, Bin Chen and Akihiro Yamazaki gave useful comments.

**Competing interests.** The authors declare that they have no conflict of interest.

**Acknowledgements**. This work was supported by the National Key Research and Development Program of China (2017YFB0503603) and the National Natural Science Foundation of China (41975178, 41825011, 41301381, 41475026 and 41705014).

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

**Table 1. Averaged volume size distribution in each bin and the differences between the two versions at Qionghai site during February 2013 to December 2015.**

| Radius (μm) | Mean_ VOL_V5.0 | Std_ VOL_V5.0 | Mean_ VOL_V4.2 | Std_ VOL_V4.2 | Mean_ $\Delta$VOL | Std_ $\Delta$VOL | Mean_$\Delta$VOL/ VOL_V4.2 (%) |
|---|---|---|---|---|---|---|---|
| 0.0121 | 0.00002 | 0.00002 | 0.00009 | 0.00053 | -0.00006 | 0.00052 | -66.67 |
| 0.0177 | 0.00020 | 0.00017 | 0.00040 | 0.00217 | -0.00020 | 0.00208 | -50.00 |
| 0.0259 | 0.00104 | 0.00093 | 0.00116 | 0.00471 | -0.00012 | 0.00426 | -10.34 |
| 0.0378 | 0.00337 | 0.00290 | 0.00316 | 0.00651 | 0.00021 | 0.00580 | 6.65 |
| 0.0553 | 0.00821 | 0.00634 | 0.00970 | 0.00799 | -0.00150 | 0.00849 | -15.46 |
| 0.0809 | 0.01924 | 0.01304 | 0.02574 | 0.01216 | -0.00651 | 0.01308 | -25.29 |
| 0.1183 | 0.03959 | 0.02213 | 0.04698 | 0.02060 | -0.00739 | 0.01440 | -15.73 |
| 0.1729 | 0.05205 | 0.02655 | 0.05365 | 0.02762 | -0.00160 | 0.00858 | -2.98 |
| 0.2529 | 0.03683 | 0.02030 | 0.03704 | 0.02110 | -0.00021 | 0.00558 | -0.57 |
| 0.3698 | 0.01625 | 0.01120 | 0.01726 | 0.01155 | -0.00101 | 0.00407 | -5.85 |
| 0.5408 | 0.01046 | 0.01178 | 0.00989 | 0.01048 | 0.00058 | 0.00383 | 5.86 |
| 0.7908 | 0.01309 | 0.01217 | 0.01242 | 0.01090 | 0.00067 | 0.00535 | 5.39 |
| 1.1560 | 0.01942 | 0.01305 | 0.01952 | 0.01483 | -0.00010 | 0.00753 | -0.51 |
| 1.6910 | 0.02766 | 0.01786 | 0.02707 | 0.02105 | 0.00059 | 0.00913 | 2.18 |
| 2.4730 | 0.03812 | 0.03021 | 0.03391 | 0.03145 | 0.00421 | 0.01005 | 12.42 |
| 3.6170 | 0.05222 | 0.05550 | 0.04139 | 0.04655 | 0.01083 | 0.02102 | 26.17 |
| 5.2890 | 0.05585 | 0.07943 | 0.05122 | 0.06994 | 0.00463 | 0.02892 | 9.04 |
| 7.7340 | 0.03647 | 0.06242 | 0.06948 | 0.12479 | -0.03301 | 0.07272 | -47.51 |
| 11.3100 | 0.01348 | 0.02238 | 0.09903 | 0.24624 | -0.08555 | 0.22908 | -86.39 |
| 16.5400 | 0.00329 | 0.00400 | 0.10389 | 0.31131 | -0.10061 | 0.30881 | -96.84 |

$\Delta$VOL was defined as: $\Delta$VOL= VOL_V5.0 - VOL_V4.2;

**Table 2. The same as Table 1 but for Yucheng during February 2013 to December 2015.**

| Radius (μm) | Mean_ VOL_V5.0 | Std_ VOL_V5.0 | Mean_ VOL_V4.2 | Std_ VOL_V4.2 | Mean_ $\Delta$VOL | Std_ $\Delta$VOL | Mean_$\Delta$VOL/ VOL_V4.2 (%) |
|---|---|---|---|---|---|---|---|
| 0.0121 | 0.00004 | 0.00003 | 0.00009 | 0.00057 | -0.00005 | 0.00057 | -55.56 |
| 0.0177 | 0.00037 | 0.00034 | 0.00041 | 0.00226 | -0.00005 | 0.00227 | -12.20 |
| 0.0259 | 0.00209 | 0.00223 | 0.00127 | 0.00466 | 0.00082 | 0.00509 | 64.57 |
| 0.0378 | 0.00771 | 0.00814 | 0.00365 | 0.00668 | 0.00405 | 0.01027 | 110.96 |
| 0.0553 | 0.01964 | 0.01793 | 0.01092 | 0.01085 | 0.00872 | 0.0201 | 79.85 |
| 0.0809 | 0.03693 | 0.02529 | 0.02626 | 0.01986 | 0.01067 | 0.02768 | 40.63 |
| 0.1183 | 0.05215 | 0.02732 | 0.04296 | 0.02867 | 0.00919 | 0.02261 | 21.39 |
| 0.1729 | 0.05234 | 0.02986 | 0.04693 | 0.03068 | 0.00541 | 0.00922 | 11.53 |
| 0.2529 | 0.03565 | 0.02656 | 0.03577 | 0.02494 | -0.00012 | 0.00731 | -0.34 |
| 0.3698 | 0.02127 | 0.02186 | 0.0224 | 0.01984 | -0.00112 | 0.00647 | -5.00 |
| 0.5408 | 0.02097 | 0.0232 | 0.01849 | 0.01942 | 0.00248 | 0.00739 | 13.41 |
| 0.7908 | 0.03422 | 0.02713 | 0.02756 | 0.02358 | 0.00665 | 0.00851 | 24.13 |
| 1.156 | 0.05671 | 0.04448 | 0.04648 | 0.03793 | 0.01023 | 0.01674 | 22.01 |
| 1.691 | 0.07164 | 0.05576 | 0.06556 | 0.05141 | 0.00608 | 0.02186 | 9.27 |
| 2.473 | 0.08896 | 0.06535 | 0.08213 | 0.06165 | 0.00684 | 0.01965 | 8.33 |
| 3.617 | 0.12994 | 0.10367 | 0.10484 | 0.07932 | 0.02511 | 0.03979 | 23.95 |
| 5.289 | 0.13706 | 0.12457 | 0.13461 | 0.10872 | 0.00246 | 0.05025 | 1.83 |
| 7.734 | 0.08079 | 0.09025 | 0.16487 | 0.17992 | -0.08408 | 0.1097 | -51.00 |
| 11.31 | 0.02645 | 0.03229 | 0.18977 | 0.34336 | -0.16332 | 0.31962 | -86.06 |
| 16.54 | 0.00575 | 0.00592 | 0.16934 | 0.42522 | -0.16359 | 0.42143 | -96.60 |

$\Delta$VOL was defined as: $\Delta$VOL= VOL_V5.0 - VOL_V4.2

**Table 3.** Averaged single scattering albedo and refractive index in SKYRAD 5.0 and 4.2, and the absolute and percentage differences between the two versions at Qionghai site during February 2013 to December 2015.

| | 400 nm | 500 nm | 670 nm | 870 nm | 1020 nm |
|---|---|---|---|---|---|
| $\omega_{v5.0}$ | 0.8852 | 0.9233 | 0.9355 | 0.9447 | 0.8774 |
| $\omega_{v4.2}$ | 0.8862 | 0.9260 | 0.9427 | 0.9524 | 0.8735 |
| $m_{r\_v5.0}$ | 1.4498 | 1.4423 | 1.4402 | 1.4497 | 1.5103 |
| $m_{r\_v4.2}$ | 1.4139 | 1.4305 | 1.4421 | 1.4613 | 1.4626 |
| $m_{i\_v5.0}$ | 0.0156 | 0.0086 | 0.0062 | 0.0049 | 0.0141 |
| $m_{i\_v4.2}$ | 0.0137 | 0.0077 | 0.0047 | 0.0034 | 0.0120 |
| $\delta\omega$ | -0.0009 | -0.0028 | -0.0072 | -0.0077 | 0.0039 |
| $\sigma(\delta\omega)$ | 0.0268 | 0.0287 | 0.0283 | 0.0332 | 0.0454 |
| $\delta m_r$ | 0.0359 | 0.0118 | -0.0019 | -0.0116 | 0.0477 |
| $\sigma(m_r)$ | 0.0484 | 0.0413 | 0.0380 | 0.0441 | 0.0740 |
| $\delta m_i$ | 0.0019 | 0.0009 | 0.0015 | 0.0016 | 0.0021 |
| $\sigma(m_i)$ | 0.0038 | 0.0039 | 0.0038 | 0.0041 | 0.0054 |
| $\delta\omega\%$ | -0.1057 | -0.2984 | -0.7596 | -0.8090 | 0.4443 |
| $\delta m_r\%$ | 2.5363 | 0.8263 | -0.1321 | -0.7926 | 3.2598 |
| $R_{mi}$ | 1.1374 | 1.1175 | 1.3298 | 1.4694 | 1.1707 |

$\omega$, $m_r$ and $m_i$ mean averaged single scattering albedo, real part of refractive index and the imaginary part of refractive index; subscript v5.0 and v4.2 mean parameters retrieved by SKYRAD V5.0 and V4.2, respectively; $\delta$- and $\delta$-% mean absolute and percentage difference between SKYRAD V5.0 and V4.2, respectively; $\sigma(\delta)$ mean the standard deviation of absolute differences between SKYRAD V5.0 and V4.2; $R_{mi}$ means the ratio of $m_{i\_v5.0}$ to $m_{i\_v4.2}$.

**Table 4.** The same as Table 3 but for Yucheng during February 2013 to December 2015.

| | 400 nm | 500 nm | 670 nm | 870 nm | 1020 nm |
|---|---|---|---|---|---|
| $\omega_{v5.0}$ | 0.8944 | 0.9343 | 0.9352 | 0.9390 | 0.9022 |
| $\omega_{v4.2}$ | 0.9086 | 0.9284 | 0.9360 | 0.9454 | 0.9122 |
| $m_{r\_v5.0}$ | 1.4535 | 1.4590 | 1.4768 | 1.5115 | 1.5404 |
| $m_{r\_v4.2}$ | 1.4950 | 1.5055 | 1.5089 | 1.5409 | 1.5358 |
| $m_{i\_v5.0}$ | 0.0100 | 0.0053 | 0.0054 | 0.0053 | 0.0093 |
| $m_{i\_v4.2}$ | 0.0085 | 0.0058 | 0.0047 | 0.0039 | 0.0076 |
| $\delta\omega$ | -0.0142 | 0.0059 | -0.0008 | -0.0064 | -0.0101 |
| $\sigma(\delta\omega)$ | 0.0188 | 0.0180 | 0.0208 | 0.0267 | 0.0421 |
| $\delta m_r$ | -0.0415 | -0.0465 | -0.0322 | -0.0294 | 0.0045 |
| $\sigma(m_r)$ | 0.0631 | 0.0506 | 0.0448 | 0.0520 | 0.0609 |
| $\delta m_i$ | 0.0015 | -0.0005 | 0.0007 | 0.0013 | 0.0017 |
| $\sigma(m_i)$ | 0.0030 | 0.0029 | 0.0030 | 0.0035 | 0.0048 |
| $\delta\omega\%$ | -1.5646 | 0.6408 | -0.0873 | -0.6766 | -1.1048 |
| $\delta m_{r\%}$ | -2.7743 | -3.0869 | -2.1317 | -1.9053 | 0.2960 |
| $R_{mi}$ | 1.1798 | 0.9099 | 1.1455 | 1.3383 | 1.2230 |

$\omega$, $m_r$ and $m_i$ mean averaged single scattering albedo, real part of refractive index and the imaginary part of refractive index; subscript v5.0 and v4.2 mean parameters retrieved by SKYRAD V5.0 and V4.2, respectively; $\delta$- and $\delta$-% mean absolute and percentage difference between SKYRAD V5.0 and V4.2, respectively; $\sigma$ ($\delta$) mean the standard deviation of absolute differences between SKYRAD V5.0 and V4.2; $R_{mi}$ means the ratio of $m_{i\_v5.0}$ to $m_{i\_v4.2}$.