# Peer review of "Inter-comparison between the Aerosol Optical Properties Retrieved by Different Inversion Methods from SKYNET Sky Radiometer Observations over Qionghai and Yucheng in China"

_Atmospheric Measurement Techniques, 2019_

## Referee Comment (RC1) · Anonymous Referee #2 · 1 Apr 2019

The present study is highly important for both SKYNET products and for other sunphotometric instruments and networks that could potentially benefit from the enhanced approached of SKYRAD inversions methodology. A detailed comparison between the two versions of SKYRAD is missing from the literature and it is always a question for scientists handling SKYNET data. Stations selected for the study seem to provide a sufficient amount of data for this comparison. Additionally, authors have exploited these datasets to provide a climatology of aerosol properties at both measuring locations. However, the manuscript lacks of explanations on the causes of differences between

the v4.2 and v5.0 retrievals and sufficient evidence on the actual seasonal variability of aerosols in the two regions. Algorithms of both versions are clearly described, but it is crucial to pinpoint and discuss the way the differences between the versions affect the retrievals. Since the two algorithms are not treated as a "black box", it should be more clear which physical processes affects the retrievals and which atmospheric conditions could lead to highest uncertainty. At least some discussion on the uncertainty of each variable in each approach should be provided. Also, the part about the seasonal variability of aerosols properties, results are presented but not investigated and discussed in the level expected for a scientific study. Majority of readers are not unfamiliar with local weather systems and patterns, emissions, and these should be considered for publication in AMT after a major revision addressing these concerns.

**Specific comments**

P2 I5-7 This should be divided in two sentences because it is confusing. P3 I17 I assume this precision is for the sky radiance measurement, but it should restated to be clear. P3 I18 Some details on the calibration of these instruments should be added. P3 I19-24 More detailed description of the locations is needed. P5 I10 More details on the quality control and cloud screening procedures should be provided. P5 I5 Since the algorithm uses a priori a bimodal SDF, it should be presented as a finding that the retrieved SDF is bimodal (in abstract, conclusions and discussion of seasonal variations) P5 I17. I It would be useful to report the number of measurements fulfilling this criterion at both sites P5 I27 This sentence indicates that v 5.0 is more erroneous in coarse mode. Is there more evidence on that? Is that strictly due to algorithmic reasons? More discussion is needed on this effect.

Figure 1. It is really difficult to visually distinguish the differences between the two versions for most bins. Probably a different approach should be also demonstrated here (absolute differences, relative differences? histogram?) to facilitate reader's comprehension. Also a x axes label is missing.
Paragraph 3.2 Some physical interpretation and discussion about these differences is missing. Are they explained strictly algorithmically or is there some natural process driving them? Differences of SSA are really high, and have opposite behavior (more absorbing for v 5.0 in Qionghai and more scattering in Yucheng). Keep in mind that SSA values in the atmosphere have a very small range, and these differences are very high. SSA at 0.92 and 0.86 (example at Qionghai at January) indicate totally different types of aerosol. In addition, the different behavior at the two sites, makes it difficult to assume some systematical bias. Since there are no other independent data to validate which version is closest to the actual condition, I strongly suggest to investigate further this behavior. In the scientific literature you could find a number of approaches to select depending on the data available, but it is crucial at this point to have some evidence on the validity of the retrievals.

Figure 2. bar plots for mean monthly values and showing only the higher part of the error bar (which I assume is standard deviation but nowhere stated) is confusing. I suggest to visualize in another way. P7 I17 Frequency distributions are plotted in figure 3. Where probability distributions mentioned here could be found?

Figure 3. x axis label is missing Figure 4. x axis label is missing

General comment for 3.2-3.3. First, a more uniformly approach on the presentation of results should be applied. Treating histograms for refractive index and monthly averages for ssa, makes the datasets incomprehensible, since cannot be easily combined and provide a conclusion on the behavior. Also, some conclusions should be reached linked to the differences of the two versions and the causes of the variations. For that purpose, there should be some discussion about the algorithmic differences and the outputs. Finally, It is important to understand whether other parameters are linked to the differences. At least it should be investigated the corresponding aerosol loads (AOD) for each case. Does the difference increase/decrease with higher AOD? Is the elevation of the sky radiance measurements linked to the differences between the two versions?

AMTD
P9 I16-17 Some reference or some data are needed to provide evidence about the meteorological argument.

P9 I20 Further discussion and evidence are needed to support this argument.

P10 I14. By definition, SSA will decrease when absorbing aerosols increase. This sentence does not provide any explanation on the behavior. More detailed discussion should be added on these results.

Paragraph 3.4.2 Since for SSA the selection of version 4.2 or 5.0 could lead to different conclusions on the type of aerosols, some discussion on that issue should be added here. P10 I26-17 Since the algorithm uses bimodal fits, there was no way to find a different distribution

P11 I2-4. Why anthropogenic aerosols should decrease in winter/spring ? Are there any information on the human activities in the area? Why sea salt aerosols increase? Also, some information about the monsoonal influence in the region should be added for the readers that are unfamiliar with local climatology (preferably at the site description section at page 3)

P11 I6-8 Also fine mode is very high in summer (compared to Qionghai). Any interpretation on that? The only source of large particles in the area is dust long transport or are there any other sources? P11 I8. Also fine mode seemed to peak to almost double values in summer/winter compared to spring/autumn. Is there are any explanation for this behavior?

Figure 6. X axes label is missing

General comment for 3.4 I suggest to summarize the types and variations of aerosols in both sites in a more descriptive way at the end. Also, It would be very useful a discussion –based on earlier paragraphs- on the properties and conditions that both version come together and the conclusions that have higher uncertainties due to the deviations between the algorithms.
P12 I16-18 There is no evidence in the study of the cause of this behavior (algorithm or type of aerosols?). More work should be done before coming to this conclusion.

---

## Referee Comment (RC2) · Anonymous Referee #1 · 2 Apr 2019

This analysis is divided in two parts: first, a comparison of aerosol properties retrieved by inversion code SKYRAD versions 4.2 and 5.0 is performed, based on two years of data for two SKYNET sites. Second, version 5.0 is used to analyze the aerosol characteristics at the two sites. This kind of study is needed for the improvement of the SKYNET network methodology, and also for the improvement of our knowledge of the aerosol characteristics at China. Therefore, it is adequate for this journal.

However, I would recommend to accept the paper after a major revision, mainly related to: - adding detail to the text - improving the graphical representations - further

discussing the temporal behavior of aerosol at the two sites

The use of English is adequate, although some flaws are pointed out and I would recommend a revision.

General comments: - Introduction: Some more background discussion would be welcome. Please add also a few comments about general differences between versions 4.2 and 5.0. Non-sphericity, minimization technique used, etc, so general readers can learn about these codes. - Section 2.1: PLease explain the method used for the calibration, and any findings you consider interesting to note, if any (calibration drift, etc). It is important also to detail the description of the two sites, including a map if possible, to understand aerosol characteristics. - Section 2.2: Please cite the source for the details given about version 5.0. Comments about expected errors would be useful at this stage. - Section 3.1: It is possible to further analyze the comparison of the SDF, including some statistics. In the first part of the paper, perhaps the authors should focus on the analysis of the differences (absolute or relative) and leave the absolute retrievals for the second part of the study (analysis of the aerosol properties). Why AOD is not included in the comparison? - Section 3.2. and 3.3.: similarly yo 3.1, concentrate on differences rather than absolute values. Finally, add your opinion about the most adequate version to use in the remaining, based on the results, so both parts of paper are smoothly linked. - Section 3.4: I think the analysis of the aerosol properties at the two sites need a deeper analysis, also including references to previous analysis from China or elsewhere. Line 251 is particularly vague, as other reasons for the increase of AOD in summer are usually considered (differences in transport from remote areas, increase of secondary aerosols due to higher solar radiation...). In contrast to first part of the paper, in the second part I would recommend to focus on the absolute values, represented in monthly means along the year, with corresponding boxplots, for example. Current analysis based on seasonal averages alone, is not optimum.

Other specific corrections: - line 59: many -> several? - line 74: There are a few - line 94: The dynamic range seems should be 10ˆ7 instead of 107? - line 120-121: rewrite

(parenthesis?) - line 131-132: eˆ2 - line 156: more comments on the cloud sccreening and quality control - line 173: it is important to highlight the fact that the unrealistic coarse mode in v4.2 is removed - line 244: The AOD is - line 289-291: three significant digits is enough for the refractive index (1.45 etc)

---

## Author Comment (AC2) · 7 Sep 2019

We appreciate the reviewer' valuable comments and constructive suggestions which help us improve the quality of the manuscript. We have carefully revised the manuscript according to these comments. Point-to-point responses are provided below. The reviewer' comments are in black, our responses are in blue and the corresponding changes in manuscript are in red.

Reviewer #2

This analysis is divided in two parts: first, a comparison of aerosol properties retrieved by inversion code SKYRAD versions 4.2 and 5.0 is performed, based on two years of data for two SKYNET sites. Second, version 5.0 is used to analyze the aerosol characteristics at the two sites. This kind of study is needed for the improvement of the SKYNET network methodology, and also for the improvement of our knowledge of the aerosol characteristics at China. Therefore, it is adequate for this journal.

However, I would recommend to accept the paper after a major revision, mainly related to: adding detail to the text - improving the graphical representations - further discussing the temporal behavior of aerosol at the two sites

The use of English is adequate, although some flaws are pointed out and I would recommend a revision.

**Response:** Thank you for your valuable comments and constructive suggestions. We have carried out additional experiments and found that the calibration constants in the previous experiments were incorrect, so we corrected them and re-carried out the experiments and numerical tests; some results and figures have been updated and represented in the following response.

We have added more details related to the graphical, climate and major chemical compositions in PM2.5 in the two sites. Some of the new figures and comments about inter-comparisons results between V5.0 and V4.2 have been shown in the following comments. Meanwhile, we have also investigated some parameters linked to the SSA differences between the V5.0 and V4.2, the seasonal variation of aerosol have been discussed combining the possible emission sources and prevailing wind based on more data and references as shown in the following comments.

**General comments:**

Introduction: Some more background discussion would be welcome. Please add also a few comments about general differences between versions 4.2 and 5.0. Non-sphericity, minimization

technique used, etc, so general readers can learn about these codes.

**Response:** Non-sphericity particle model are neither included in V4.2 nor V5.0. We have added the comment in the revised manuscript.

V4.2 uses the iterative relaxation method of Nakajima et al. (1983, 1996) to remove the multiple scattering contributions and derive an optimal solution using a statistical regularization method (Turchin and Nozik, 1969) by minimizing the cost function as proposed by Phillips (1962) and Twomey (1963).

V5.0 uses the non-linear maximum likelihood method defined by Rodgers (2000) which was based on the Bayesian theory. The non-linear inversion has a strong dependence on the estimation of the first-guess solution. Version 5.0 uses an a priori SDF of a bimodal log-normal function.

Section 2.1: Please explain the method used for the calibration, and any findings you consider interesting to note, if any (calibration drift, etc).

**Response:** The Improved Langley (IL) plot method is used in this study to determine the temporal and spectral calibration constants for direct intensity (F0) with accuracy of about 1.0–2.5 %, depending on the wavelength (Nakajima et al., 1996; Campanelli et al., 2004). The calibration by IL plot method is made daily, the variation of F0 due to instrumental drift can be quickly spotted, and then appropriate corrections to data can be applied exactly from the period in which the deviation occurred (Campanelli et al., 2004). We have added the above comments in the revised manuscript.

It is important also to detail the description of the two sites, including a map if possible, to understand aerosol characteristics. -

**Response:** We have added a map in the revised manuscript to show the locations of the two SKYNET sites in this study as shown in the following figure.

[Figure]

**Figure 1: The locations of the two SKYNET sites in the study**

In section 2.2 Site description, we have added some more details including monsoon, temperature, and precipitation. We have added the following descriptions in the revised manuscript.

The Qionghai site of SKYNET (19.23°N, 110.46°E, 24 m a.s.l.), which was located in the eastern part of Hainan Island, was mainly influenced by East Asia monsoons and typhoons. During summer, the dominant wind is from south to southeast, summer monsoon from the South China Sea and West Pacific brought most of the annual rainfall to the island (Zhu et al., 2005), whereas the winter monsoon from Inner Mongolia carries dry winds to the area (Zhu et al., 2005; Peel et al., 2007; Yin et al., 2002). Annual average rainfall in Qionghai is estimated about 1653.4 mm. Maximum high temperature occurs in July, with monthly average of 28.6 °C, monthly lowest temperature occurs in January, with monthly average of 19.1°C (Yin et al., 2002).

The other measurement site in this study was located in rural Yucheng (36.82°N, 116.57°E, 22 m a.s.l.), Shandong Province, China, which is almost in the centre of the North China Plain. The selected site is in an open field surrounded by farmland. The region belongs to semi-humid and temperate monsoon climate zone, characterized by a mean annual temperature of 21°C and mean annual precipitation of 610 mm mainly distributed in summer months (Chen et al., 2012).Yucheng and the surrounding areas are famous for their agriculture (e.g., wheat and corn) and grazing land (e.g., donkeys and chickens). In addition, the site near 20 to 30 km radius located several factories in the production of inorganic and organic fertilizers (Wen et al., 2015), and the application of fertilisers to farmland emitted a great deal of NH3 (Zhao et al., 2012). Meanwhile, Yucheng was located in the downwind of the Beijing-Tianjin-Hebei region, long-distance transport of sources of industrial

pollution and biomass burning contributed significantly to the concentrations of pollutants in Yucheng (Lu et al., 2016).

In addition, based on the results simulated by the Community Multi-scale Air Quality model with the 2D Volatility Basis Set (CMAQ/2D-VBS) (23), we have added the following comments and figure in the revised manuscript to describe the major chemical compositions in PM2.5 and their percentage contribution to PM2.5 at the two sites.

It is well known that OC, EC, $SO_4^{2-}$, $NO3^-$ and $NH4^+$ were the dominant chemical components in PM2.5 (Tao et al., 2017). The above-mentioned five major components over the two sites were discussed below based on the results simulated by the Community Multi-scale Air Quality model with the 2D Volatility Basis Set (CMAQ/2D-VBS) (23) at 36- × 36-km resolution with emission inputs derived from a Chinese emission inventory developed and updated to 2015 with details in these studies (Wang et al., 2014; Zhao et al., 2018). The contributors to carbonaceous aerosols in China mainly include coal combustion, vehicle exhaust and biomass burning, etc (Liu et al., 2018).As shown in Fig. 9, the concentrations of OC were significantly higher than that of EC at Qionghai, likely due to the mixed contributions of atmospheric chemical reactions and primary anthropogenic sources to OC (Cao et al., 2004). The nitrate accounted for a large fraction of PM2.5 in Yucheng, it was strongly related to the high emission levels of NH3 and O3 in Yucheng (Wen et al., 2015).

[Figure]

**Figure 9: Percentage (%) contribution of $NO3^-$, $SO4^{2-}$,$NH4^+$ ,OC and EC to PM2.5 mass in Qionghai (a)and Yucheng (b) in 2015**

Section 2.2: Please cite the source for the details given about version 5.0. Comments about expected errors would be useful at this stage.

**Response:** The non-linear maximum likelihood method used in V5.0 has a strong dependence on the estimation of the first-guess solution. Version 5.0 uses an a priori SDF of a bimodal

log-normal function. The reference 'Hashimoto et al., 2012' gives more details about V5.0. They had performed various test simulations with SKYRAD.pack V4.2 and V5.0 (Hashimoto et al., 2012), and found: In the case of a large amount of coarse particles with radius greater than 10 μm existing , the numerical tests performed by Hashimoto et al showed that V4.2 could retrieve the SDF relatively well, including the coarse mode, in comparison with V5.0, because the smoothness condition given by Eq. (2) allowed the retrieved SDF to be distributed beyond 10 μm radius, on the other hand, V5.0 underestimated the coarse mode of the SDF because of the strong SDF constraint condition given by Eq. (5) with a small model radius $rm_2$ = 2.0 μm for the coarse mode SDF (Hashimoto et al., 2012). So we have compared the differences between retrieved SSAs at 500 nm byV5.0 and V4.2 when set rm2 = 1.5, 1.8, 2.0(default), 2.5 and 3.0 in Skyrad.pack V5.0 based on the measurements in 2014. As shown in Fig.7, SSAs by V5.0 correlated to SSAs by V4.2 with R= 0.860, 0.837, 0.855, 0.809 and 0.226 when $r_{m2}$ = 1.5, 1.8, 2.5 and 3.0 in V5.0 over Qionghai, respectively. The correlation coefficient between SSA by V5.0 and V4.2 was the highest while setting $r_{m2}$ as 2.0 (the default value) at both the two sites.

[Figure]

**Figure 7: Scattergrams of retrieved SSA between SKYRAD V4.2 and V5.0 when $r_{m2}$ =2.0(default), 1.5, 1.8, 2.5 and 3.0 for Qionghai (a) and Yucheng (b) in 2014. $r_{m2}$ represents the model radius for the coarse mode SDF.**

We also investigated whether the total amount of aerosols in the atmosphere were linked to the difference in SSA between the two versions. As shown in Fig. 8, the SSA differences at 500nm between the two versions (defined as: |SSA_V5.0$_{500nm}$ - SSA_V4.2$_{500nm}$ |) decreased while the corresponding AODs at wavelengths of 500 nm by V5.0 increased at both the two sites. When the

AOD was high (in this study the threshold was set to 0.5 for $AOD_{500nm}$), SSA retrieved by V5.0 had a good comparison with those with V4.2. It is well known that the inversion products have a very high uncertainty in cases of very low aerosol burdens, the retrieval error in SSA rapidly increases with decreasing AOD (Dubovik et al., 2000), especially in parameters such as the imaginary part of the refractive index.

We have added the above comments and figure in the revised manuscript.

Section 3.1: It is possible to further analyze the comparison of the SDF, including some statistics. In the first part of the paper, perhaps the authors should focus on the analysis of the differences (absolute or relative) and leave the absolute retrievals for the second part of the study (analysis of the aerosol properties). Why AOD is not included in the comparison?

**Response:** Following the reviewer's suggestion, we have added the following figure which showed the plots of AOD values at each wavelength derived from the solar direct irradiance between the two versions. High correlation was found with a significant coefficient larger than 0.995 at each band except 1020nm over Qionghai. High consistency of AODs between V4.2 and V5.0 indicates that the inversion process in V5.0 did not bring about a large change in the retrieved direct solar radiation (Hashimoto et al., 2012).

[Figure]

**Figure 2: Scatter plot and correspondent linear fitting for the aerosol optical depth (AOD) between SKYRAD V4.2 and V5.0 at wavelengths of 400, 500, 670, 870, and 1020 nm over Qionghai and Yucheng during February 2013 to December 2015.**

We have added the above comments and figure in the revised manuscript.

We have replaced Fig.1 with the following figure which shows the retrieved monthly volume size distribution between SKYRAD V4.2 (red lines) and V5.0 (blue lines) for Qionghai (dotted line) and Yucheng (solid lines) during February 2013 to December 2015. As shown in the following figure, V4.2 showed a tri-model pattern with three peak volume at radius of 0.25, 1.16, 11.31 and 0.25, 1.69, 11.31 in July and September over Yucheng, respectively. Figure 3 also showed that there were larger differences in volume SDF of the coarse mode between V4.2 and V5.0 at Qionghai than those at Yucheng in most months.

[Figure]

**Figure 3: Retrieved monthly volume size distribution between SKYRAD V4.2 (red lines) and V5.0 (blue lines) for Qionghai (dotted line) and Yucheng (solid lines) during February 2013 to December 2015**

We have added the above figure and comments in the revised manuscript.

Section 3.2. and 3.3.: similarly to 3.1, concentrate on differences rather than absolute values. Finally, add your opinion about the most adequate version to use in the remaining, based on the results, so both parts of paper are smoothly linked.

**Response:** Based on the new experiment results as shown in the following figures, the SSA and $m_i$ had relatively high correlation coefficients between V4.2 and V5.0 with default $rm_2$ value based on the above comparison results. In addition, some tests by Hashimoto et al showed that the SDF setting in V5.0 was useful for detecting ill-conditioned data caused by cirrus contaminations, horizontally and/or temporally inhomogeneous aerosol stratification, and so on (Hashimoto et al., 2012). So we still chose the retrieved results by V5.0 to analyze the seasonal variability of the

aerosol optical properties over Qionghai and Yucheng.

[Figure]

**Figure 4: Scattergrams of the single scattering albedo between SKYRAD 4.2 and 5.0 at wavelengths of 400, 500, 670, 870, and 1020 nm over Qionghai and Yucheng during February 2013 to December 2015. Only data with $AOD_{500nm}>0.2$ are shown. The green line means the fitted linear regression curve.**

[Figure]

**Figure 5: Scattergrams of the imaginary part of the complex refractive index ($m_i$) results between SKYRAD 4.2 and 5.0 at wavelengths of 400, 500, 670, 870, and 1020 nm over Qionghai and Yucheng during February 2013 to December 2015.**

We have added the above comments in the revised manuscript.

Section 3.4: I think the analysis of the aerosol properties at the two sites need a deeper analysis, also including references to previous analysis from China or elsewhere.

**Response:** Based on the new experiment results, we have made major revision on this section. Section 3.1, 3.2 , 3.3 have been merged into Section 3.1, Section 3.4.1, 3.4.2, 3.4.3, 3.4.4 have been changed into Section 3.2.2, 3.2.3, 3.2.4 , 3.2.5 as follows, Section 3.2.1 is 'The major chemical compositions in PM2.5 at the two sites' as the above, the changes in the manuscript are in red.

**3.2.2 AOD**

[revised manuscript text omitted]

Qionghai and Yucheng. The averages of $m_r$ at 500 nm in Qionghai were 1.45, 1.46, 1.45, and 1.43 in

spring, summer, fall, and winter, respectively. The averages of the real parts were higher in spring

compared to the other seasons in Yucheng. The $m_r$ in Yucheng showed a maximum of approximately

1.47 in spring and a minimum of approximately 1.45 in summer.

Figure 12b presented the seasonal variation of the imaginary part of the refractive index at 500 nm over

Qionghai and Yucheng. On the contrary to SSA, the results of imaginary part of complex refractive

index ($m_i$) were both highest in winter in the two sites.

Aerosol absorption coefficient was determined by elemental carbon (EC) mass concentration and its

coating (Tao et al., 2017), heating activities and biomass burning induced higher carbonaceous aerosols

in winter in Yucheng. As shown in Fig.9, OC/EC ratios could be estimated to be greater than 2.0 in

Qinghai, suggesting coal and vehicle exhaust as dominant carbonaceous aerosols sources (He et al.,

2008; Watson et al., 2001).

[Figure]

**Figure 12: Seasonal variations in the real part of the refractive index (a) and the imaginary part of the**

**refractive index (b) based on SKYRAD V5.0 over Qionghai and Yucheng for the period from February 2013**

**to December 2015. The boxes represent the 25th to 75th percentiles of the distributions while the dots and**

**solid lines within each box represent the means and medians, respectively.**

Line 251 is particularly vague, as other reasons for the increase of AOD in summer are usually

considered (differences in transport from remote areas, increase of secondary aerosols due to

higher solar radiation...). In contrast to first part of the paper, in the second part I would

recommend to focus on the absolute values, represented in monthly means along the year, with

corresponding boxplots, for example. Current analysis based on seasonal averages alone, is not optimum.

**Response:**

The increase of AOD in summer in Yucheng maybe was caused by hygroscopic effects. Yucheng and the surrounding areas are famous for their agriculture and grazing land. In addition, the site near 20 to 30 km radius located several factories in the production of inorganic and organic fertilizers (Wen et al., 2015), and the application of fertilisers to farmland emitted a great deal of NH3 in summer (Zhao et al., 2012). The humidity of Yucheng (belong to Shandong province) is highest in summer than other seasons (Meng et al., 2007). High humidity combined with large fractions of hygroscopic chemical components can enhance light extinction and haze intensity the scattering coefficient of secondary inorganic aerosols (such as sulfate, nitrate and ammonium) (Tao et al., 2017).

We have tried to represent the monthly means with corresponding boxplots as follows. The references to previous analysis from China or elsewhere related to the two sites are mostly based on seasonal averages. There are no other data available to be inter-compared with our results during the experiment time. To analyze the reasons for the variation or inter-compare based on the references, the temporal variation analysis are based on seasonal averages as above .

[Figure]

**Figure : Monthly variations in the AOD based on SKYRAD V5.0 over Qionghai (a) and Yucheng (b) for the period from February 2013 to December 2015. The boxes represent the 25th to 75th percentiles of the distributions while the dots and solid lines within each box represent the means and medians, respectively.**

[Figure]

**Figure : Monthly variations in the single scattering albedo (SSA) based on SKYRAD V5.0 over Qionghai (a) and Yucheng (b) for the period from February 2013 to December 2015. The boxes represent the 25th to 75th percentiles of the distributions while the dots and solid lines within each box represent the means and medians, respectively.**

[Figure]

**Figure : Monthly variations in the imaginary part of the refractive index based on SKYRAD V5.0 over Qionghai (a) and Yucheng (b) for the period from February 2013 to December 2015. The boxes represent the 25th to 75th percentiles of the distributions while the dots and solid lines within each box represent the means and medians, respectively.**

**Other specific corrections:**

line 59: many -> several? -

**Response:** We have replaced "many" with "several".

line 74: There are a few -

**Response:** We have added "a" before "few".

line94: The dynamic range seems should be 10^7 instead of 107? -

**Response:** We have replaced "107" with "$10^7$".

line 120-121: rewrite (parenthesis?) -

**Response:** We missed a right parenthesis. We have added it in the revised manuscript as below.

(i.e., $\ln(r_{i+1}) - \ln(r_i) = const$)

line 131-132: e^2 -

**Response:** We have replaced "e2" with "$e^2$".

line 156: more comments on the cloud screening and quality control -

**Response:** The standard process of quality control in Skyrad.pack V4.2 and V5.0 applies a retrieval error between observations and calculated theoretical values by using retrieval values, $\sigma_{obs}$

$$\sigma_{obs} = \sqrt{W_e \sum_i \left(\frac{\tau_{\lambda_i}}{\tau_{\lambda_i}^{meas}} - 1\right)^2 + W_p \sum_i \sum_j \left[\frac{R_{\lambda_i}(\Theta_j)}{R_{\lambda_i}^{meas}(\Theta_j)} - 1\right]^2}$$

where ($\tau_{\lambda_i}^{meas}$ and $R_{\lambda_i}^{meas}$) and ($\tau_{\lambda_i}$ and $R_{\lambda_i}$) are measured and retrieved observation vectors for the AOD and relative sky radiance, $N_i$, $N_j$, and $N_{total} = N_i + N_i \times N_j$ indicate the number of measured wavelengths, scattering angles, and their total, respectively, $W_e = W_P = 1/N_{total}$. In V4.2, the data if the value of $\sigma_{obs}$ is larger than 0.2, but $\sigma_{obs}$ is set 0.07 as a threshold for data rejection in V5.0. There are some other differences between V4.2 and V5.0 on the issue of quality control of observation data and cloud screening (Hashimoto et al., 2012).

We have added the above comments in the revised manuscript.

line 173: it is important to highlight the fact that the unrealistic coarse mode in v4.2 is removed -

**Response:** The unrealistic coarse mode in v4.2 is removed in V5.0 by the constraint of a reduced SDF for particles with radius greater than 10 μm. Some tests by Hashimoto et al showed that the SDF setting in V5.0 was useful for detecting ill-conditioned data caused by cirrus contaminations, horizontally and/or temporally inhomogeneous aerosol stratification, and so on (Hashimoto et al., 2012).

We have added the above comments in the revised manuscript.

line 244: The AOD is -

**Response:** We have replaced "was" with "is".

line 289-291: three significant digits is enough for the refractive index (1.45 etc)

**Response:** Following the reviewer's suggestion, the numbers referred to the real part of the refractive index in the revise manuscript have been changed to be with three significant digits.

[revised manuscript text omitted]

---

## Author Response (AR1)

Dear Huizheng Che,

We are pleased to inform you that the open discussion of your following manuscript has been closed:

Journal: AMT
Title: Inter-comparison between the Aerosol Optical Properties Retrieved by Different Inversion Methods from SKYNET Sky Radiometer Observations over Qionghai and         Yucheng in China
Author(s): Zhe Jiang et al.
MS No.: amt-2019-39
MS Type: Research article
Special Issue: SKYNET – the international network for aerosol, clouds, and solar radiation studies and their applications (AMT/ACP inter-journal SI)

No more referee comments and short comments will be accepted. Now the public discussion shall be completed as follows:

You - as the contact author - are requested to individually respond to all referee comments (RCs) by posting final author comments on behalf of all co-authors no later than 30 May 2019 (final response phase) at: https://editor.copernicus.org/amt-2019-39/final-response

After your posts, you have to explicitly finalize the final-response form before you are asked in a separate email to prepare and submit your revised manuscript for peer-review completion and potential final publication in AMT.

When posting your author comments (ACs), you can choose between new comments or co-listing of existing ones. Please also consider replying to short comments (SCs) from the scientific community. The response to the Referees shall be structured in a clear and easy-to-follow sequence: (1) comments from Referees, (2) author's response, (3) author's changes in manuscript.

Preparation and submission of a revised manuscript for peer-review completion is encouraged only if you can satisfactorily address all comments and if the revised manuscript meets the high quality standards of AMT (https://www.atmospheric-measurement-techniques.net/peer_review/review_criteria.html). In case of doubt, please ask the handling Associate Editor directly whether they would encourage submission of a revised manuscript or not.

Please note also that the submission of a revised manuscript does not ensure publication in AMT. The Associate Editor will carefully assess your revised manuscript in view of the interactive public discussion and may forward it to the original or new Referees for further commenting.

You are invited to monitor the processing of your manuscript via your MS Overview: https://editor.copernicus.org/AMT/my_manuscript_overview

To log in, please use your Copernicus Office user ID 99680.

Thank you very much in advance for your cooperation. In case any questions arise, please do not hesitate to contact me.

Kind regards,

Natascha Töpfer
Copernicus Publications
Editorial Support
editorial@copernicus.org

on behalf of the AMT Editorial Board
* * *
We appreciate the reviewers' valuable comments and constructive suggestions which help us improve the quality of the manuscript. We have carefully revised the manuscript according to these comments. Point-to-point responses are provided below. The reviewers' comments are in black, our responses are in blue and changes in manuscript are in red.

Reviewer #1

The present study is highly important for both SKYNET products and for other sunphotometric instruments and networks that could potentially benefit from the enhanced approached of SKYRAD inversions methodology. A detailed comparison between the two versions of SKYRAD is missing from the literature and it is always a question for scientists handling SKYNET data. Stations selected for the study seem to provide a sufficient amount of data for this comparison. Additionally, authors have exploited these datasets to provide climatology of aerosol properties at both measuring locations. However, the manuscript lacks of explanations on the causes of differences between the v4.2 and v5.0 retrievals and sufficient evidence on the actual seasonal variability of aerosols in the two regions. Algorithms of both versions are clearly described, but it is crucial to pinpoint and discuss the way the differences between the versions affect the retrievals. Since the two algorithms are not treated as a "black box", it should be more clear which physical processes affects the retrievals and which atmospheric conditions could lead to highest uncertainty. At least some discussion on the uncertainty of each variable in each approach should be provided. Also, the part about the seasonal variability of aerosols properties, results are presented but not

investigated and discussed in the level expected for a scientific study. Majority of readers are not unfamiliar with local weather systems and patterns, emissions, and these should be described in the manuscript. â˜A´lThus, I suggest that the manuscript should be considered for publication in AMT after a major revision addressing these concerns.

**Response:** Thank you for your valuable comments and constructive suggestions. We have carried out additional experiments and found that the calibration constants in the previous experiments were incorrect, so we corrected them and re-carried out the experiments and numerical tests; some results and figures have been updated and represented in the following response.

We have added more details related to the graphical, climate and major chemical compositions in PM2.5 in the two sites. Some of the new figures and comments about inter-comparisons results between V5.0 and V4.2 have been shown in the following comments. Meanwhile, we have also investigated some parameters linked to the SSA differences between the V5.0 and V4.2, the seasonal variation of aerosol have been discussed combining the possible emission sources and prevailing wind based on more data and references as shown in the following comments.

**Specific comments**

P2 l5-7 This should be divided in two sentences because it is confusing.

**Response:** The sentence has been divided into two sentences in the revised manuscript. (P2 Line 9-11)

P3 l17 I assume this precision is for the sky radiance measurement, but it should restated to be clear.

**Response:** This precision is for the sky radiance measurement. To avoid confusion, we have added the following revised sentence 'The typical measurement interval of the sky radiance is 10 min' in the revised manuscript. (P3 Line 19)

P3 l18 Some details on the calibration of these instruments should be added.

**Response:** Following the reviewer's suggestion, we have added the following descriptions in the revised manuscript: Improved Langley (IL) plot method is used in this study to determine the temporal and spectral calibration constants for direct intensity (F0) with accuracy of about

1.0–2.5 %, depending on the wavelength (Nakajima et al., 1996; Campanelli et al., 2004). The calibration by IL plot method is made daily, the variation of F0 due to instrumental drift can be quickly spotted, and then appropriate corrections to data can be applied exactly from the period in which the deviation occurred (Campanelli et al., 2004). (P3 Line 20-25)

P3 l19-24 More detailed description of the locations is needed.

**Response:** We have added a map in the revised manuscript to show the locations of the two SKYNET sites in this study as shown in the following figure. (P4 Line 18)

[Figure]

**Figure 1: The locations of the two SKYNET sites in the study**

In section 2.2 Site description, we have added some more details including monsoon, temperature, and precipitation. We have added the following descriptions in the revised manuscript.

The Qionghai site of SKYNET (19.23°N, 110.46°E, 24 m a.s.l.), which was located in the eastern part of Hainan Island, was mainly influenced by East Asia monsoons and typhoons. During summer, the dominant wind is from south to southeast, summer monsoon from the South China Sea and West Pacific brought most of the annual rainfall to the island (Zhu et al., 2005), whereas the winter monsoon from Inner Mongolia carries dry winds to the area (Zhu et al., 2005; Peel et al., 2007; Yin et al., 2002). Annual average rainfall in Qionghai is estimated about 1653.4 mm. Maximum high temperature occurs in July, with monthly average of 28.6 oC, monthly lowest temperature occurs in January, with monthly average of 19.1oC (Yin et al., 2002).

The other measurement site in this study was located in rural Yucheng (36.82°N, 116.57°E, 22 m a.s.l.), Shandong Province, China, which is almost in the centre of the North China Plain. The

selected site is in an open field surrounded by farmland. The region belongs to semi-humid and temperate monsoon climate zone, characterized by a mean annual temperature of 21oC and mean annual precipitation of 610 mm mainly distributed in summer months (Chen et al., 2012).Yucheng and the surrounding areas are famous for their agriculture (e.g., wheat and corn) and grazing land (e.g., donkeys and chickens). In addition, the site near 20 to 30 km radius located several factories in the production of inorganic and organic fertilizers (Wen et al., 2015), and the application of fertilisers to farmland emitted a great deal of NH3 (Zhao et al., 2012). Meanwhile, Yucheng was located in the downwind of the Beijing-Tianjin-Hebei region, long-distance transport of sources of industrial pollution and biomass burning contributed significantly to the concentrations of pollutants in Yucheng (Lu et al., 2016). (P3 Line 27- P4 Line 16)

In addition, based on the results simulated by the Community Multi-scale Air Quality model with the 2D Volatility Basis Set (CMAQ/2D-VBS) (23), we have added the following comments and figure in the revised manuscript to describe the major chemical compositions in PM2.5 and their percentage contribution to PM2.5 at the two sites.

It is well known that OC, EC, SO42-, NO3- and NH4+ were the dominant chemical components in PM2.5 (Tao et al., 2017). The above-mentioned five major components over the two sites were discussed below based on the results simulated by the Community Multi-scale Air Quality model with the 2D Volatility Basis Set (CMAQ/2D-VBS) (23) at 36- × 36-km resolution with emission inputs derived from a Chinese emission inventory developed and updated to 2015 with details in these studies (Wang et al., 2014; Zhao et al., 2018). The contributors to carbonaceous aerosols in China mainly include coal combustion, vehicle exhaust and biomass burning, etc (Liu et al., 2018).As shown in Fig. 9, the concentrations of OC were significantly higher than that of EC at Qionghai, likely due to the mixed contributions of atmospheric chemical reactions and primary anthropogenic sources to OC (Cao et al., 2004). The nitrate accounted for a large fraction of PM2.5 in Yucheng, it was strongly related to the high emission levels of NH3 and O3 in Yucheng (Wen et al., 2015). (P13 Line 3-14)

[Figure]

**Figure 9: Percentage (%) contribution of NO3⁻, SO4²⁻,NH4⁺ ,OC and EC to PM2.5 mass in Qionghai (a)and Yucheng (b) in 2015**

P5 l10 More details on the quality control and cloud screening procedures should be provided.

**Response:** The standard process of quality control in Skyrad.pack V4.2 and V5.0 applies a retrieval error between observations and calculated theoretical values by using retrieval values, $\sigma_{obs}$

$$\sigma_{obs} = \sqrt{W_e \sum_i (\frac{\tau_{\lambda_i}}{\tau_{\lambda_i}^{meas}} - 1)^2 + W_p \sum_i \sum_j [\frac{R_{\lambda_i}(\Theta_j)}{R_{\lambda_i}^{meas}(\Theta_j)} - 1]^2}$$

where ($\tau_{\lambda_i}^{meas}$ and $R_{\lambda_i}^{meas}$) and ( $\tau_{\lambda_i}$ and $R_{\lambda_i}$ ) are measured and retrieved observation vectors for the AOD and relative sky radiance, $N_i$ , $N_j$ , and $N_{total} = N_i + N_i \times N_j$ indicate the number of measured wavelengths, scattering angles, and their total, respectively, $W_e = W_P = 1/N_{total}$. In V4.2, the data if the value of $\sigma_{obs}$ is larger than 0.2, but $\sigma_{obs}$ is set 0.07 as a threshold for data rejection in V5.0. There are some other differences between V4.2 and V5.0 on the issue of quality control of observation data and cloud screening (Hashimoto et al., 2012). We have added the above comments in the revised manuscript. (P6 Line 17- 25)

P5 l17. It would be useful to report the number of measurements fulfilling this criterion at both sites

**Response:** Thank you for your kind comments. In the new experiment, we have added the measurement data from March to December in 2015. There are 3995 measurements for 436 days fulfilling V4.2 criterion and 2159 measurements for 355 days fulfilling V5.0 criterion over Qionghai. There are 13061 measurements for 577 days fulfilling V4.2 criterion and 7921measurements for 473 days fulfilling V5.0 criterion over Yucheng.

The inter-comparisons of aerosol properties between V5.0 and V4.2 were based on 1397 measurements and 5830 measurements over Qionghai and Yucheng, respectively.

We have added the above comments in the revised manuscript. (P6 Line 29- 30)

P5 l27 This sentence indicates that v 5.0 is more erroneous in coarse mode. Is there more evidence on that? Is that strictly due to algorithmic reasons? More discussion is needed on this effect.

**Response:** In the case of a large amount of coarse particles of the dust-like aerosol type with radius greater than 10 μm existing , the numerical tests performed by Hashimoto et al (Hashimoto et al., 2012) showed that V4.2 could retrieve the SDF including the coarse mode, because the smoothness condition given by Eq. (2) in the manuscript allowed the retrieved SDF to be distributed beyond 10 μm radius, on the other hand, V5.0 underestimated the coarse mode of the SDF because of the strong SDF constraint condition given by Eq. (5) with a small model radius $rm_2 = 2.0$ μm for the coarse mode SDF (Hashimoto et al., 2012).

In the case of cirrus contamination existing, the sensitivity tests results shows that V4.2 retrieved the aerosol size distribution function (SDF) including contaminating cirrus particles larger than 10 μm, but version 5 successfully filtered out the cirrus particles by the constraint of a reduced SDF for particles with radius greater than 10 μm (Hashimoto et al., 2012).

We have revised the sentence in P5 l27 to 'The large differences in volume SDF at radius over 5 μm between V4.2 and V5.0 were mainly related to that the smoothness condition in V4.2 given by Eq. (2) allowed the retrieved SDF to be distributed beyond 10 μm radius, whereas the strong constraint on the SDF for the coarse mode particles as shown in Eq. (5) was applied in V5.0 (Hashimoto et al., 2012)'. (P7 Line 20- P8 Line 1)

Figure 1. It is really difficult to visually distinguish the differences between the two versions for most bins. Probably a different approach should be also demonstrated here (absolute differences, relative differences? histogram?) to facilitate reader's comprehension. Also a x axes label is missing.

**Response:** We have replaced Fig.1 with the following figure which shows the retrieved monthly volume size distribution between SKYRAD V4.2 (red lines) and V5.0 (blue lines) for Qionghai (dotted line) and Yucheng (solid lines) during February 2013 to December 2015. As shown in the

following figure, The SDF by V4.2 usually showed a predominant peak at the coarse mode with a radius over 10 μm. In Qionghai, the SDF by V4.2 showed a slightly tri-model pattern in February. There were tri-model patterns with three peak volumes at radius of 0.026μm, 0.25μm, 16.54μm and 0.25μm, 1.69μm, 11.31μm in volume SDF by V4.2 in August and September over Yucheng, respectively.

[Figure]

Figure 3: Retrieved monthly volume size distribution between SKYRAD V4.2 (dotted lines) and V5.0 (solid lines) for Qionghai (blue lines) and Yucheng (red lines) during February 2013 to December 2015

We have added the above figure and comments in the revised manuscript. (P7 Line13- 16)

Paragraph 3.2 Some physical interpretation and discussion about these differences is missing. Are they explained strictly algorithmically or is there some natural process driving them? Differences of SSA are really high, and have opposite behavior (more absorbing for v 5.0 in Qionghai and more scattering in Yucheng). Keep in mind that SSA values in the atmosphere have a very small range, and these differences are very high. SSA at 0.92 and 0.86 (example at Qionghai at January) indicate totally different types of aerosol. In addition, the different behavior at the two sites, makes it difficult to assume some systematical bias. Since there are no other independent data to validate which version is closest to the actual condition, I strongly suggest to investigate further this behavior. In the scientific literature you could find a number of approaches to select depending on the data available, but it is crucial at this point to have some evidence on the validity of the retrievals.

**Response:** We found that the calibration constants were incorrect in the previous experiment, so

we corrected them and re-carried out the experiments, some new results and conclusions have been got. The following figure presents the compared results of SSA between SKYRAD V4.2 and V5.0 at wavelengths of 400, 500, 670, 870, and 1020 nm over Qionghai and Yucheng during February 2013 to December 2015. SSAs by V5.0 correlated to SSAs by V4.2 with R= 0.88, 0.87, 0.90, 0.88 and 0.92 at wavelengths of 400, 500, 670, 870, and 1020 nm over Qionghai , respectively. The SSA values computed from V5.0 had correlation coefficients around 0.95, 0.95, 0.96, 0.94, 0.91 at wavelengths of 400, 500, 670, 870, and 1020 nm over Yucheng.

[Figure]

**Figure 4: Scattergrams of the single scattering albedo between SKYRAD 4.2 and 5.0 at wavelengths of 400, 500, 670, 870, and 1020 nm over Qionghai and Yucheng during February 2013 to December 2015. Only data with $AOD_{500nm}>0.2$ are shown. The green line means the fitted linear regression curve.**
We have added the above comments and figure in the revised manuscript. (P8 Line13- P9 Line7)

Figure 2. bar plots for mean monthly values and showing only the higher part of the error bar (which I assume is standard deviation but nowhere stated) is confusing. I suggest to visualize in another way.

**Response:** We have replaced Fig.2 with the scattergrams of the single scattering albedo between V4.2 and V5.0 at wavelengths of 400, 500, 670, 870, and 1020 nm over Qionghai and Yucheng during February 2013 to December 2015 as shown in the above. (P9 Line5-7)

P7 l17 Frequency distributions are plotted in figure3. Where probability distributions mentioned

here could be found?

**Response:** We have replaced Fig.3 with scattergrams of the imaginary and real part of the complex refractive index (m) results between V.2 and V5.0 as bellow. As shown in Fig. 5, the $m_i$ values by V5.0 were linearly correlated with $m_i$ by V4.2 with R=0.8947, 0.8661, 0.8658, 0.8370, 0.9131 at wavelengths of 400, 500, 675, 870 and 1020 nm in Qionghai. The correlation coefficients between $m_i$ by V5.0 and those by V4.2 at the five wavelengths were all higher than 0.89 over Yucheng. (P9 Line 10- P10 Line 12)

[Figure]

**Figure 5: Scattergrams of the imaginary part of the complex refractive index ($m_i$) results between SKYRAD 4.2 and 5.0 at wavelengths of 400, 500, 670, 870, and 1020 nm over Qionghai and Yucheng during February 2013 to December 2015. The green line means the fitted linear regression curve.**

[Figure]

**Figure 6: Scattergrams of the real part of the complex refractive index ($m_r$) results between SKYRAD 4.2 and 5.0 at wavelengths of 400, 500, 670, 870, and 1020 nm over Qionghai and Yucheng during February 2013 to December 2015. The green line means the fitted linear regression curve.**

Figure 3. x axis label is missing Figure 4. x axis label is missing

**Response:** We have replaced Fig.3 and Fig.4 with the above figures. (P10 Line5- 12)

General comment for 3.2-3.3. First, a more uniformly approach on the presentation of results should be applied. Treating histograms for refractive index and monthly averages for ssa, makes the datasets incomprehensible, since cannot be easily combined and provide a conclusion on the behavior. Also, some conclusions should be reached linked to the differences of the two versions and the causes of the variations. For that purpose, there should be some discussion about the algorithmic differences and the outputs. Finally, it is important to understand whether other parameters are linked to the differences. At least it should be investigated the corresponding aerosol loads(AOD) for each case. Does the difference increase/decrease with higher AOD? Is the elevation of the sky radiance measurements linked to the differences between the two versions?

**Response:** Based on the new experiment results, the inter-comparison results of SSA and refractive index have been all presented in the form of scatter grams as above. SSA and the imaginary part of the complex refractive index ($m_i$) from V5.0 both had higher correlation coefficients with those from V4.2 in Yucheng than in Qionghai.

V4.2 uses the iterative relaxation method of Nakajima et al. (1983, 1996) to derive the aerosol size distribution function (SDF) and other parameters, the retrieved refractive index can only be chosen from the predefined set of values.V5.0 uses the non-linear maximum likelihood method defined by Rodgers (2000) which has a strong dependence on the estimation of the first-guess solution. V5.0 used an a priori SDF of a bimodal log-normal function, we have compared the differences between retrieved SSAs at 500 nm byV5.0 and V4.2 when set $rm_2$ = 1.5, 1.8, 2.0(default), 2.5 and 3.0 in Skyrad.pack V5.0 based on the measurements in 2014. As shown in the following figure, SSAs by V5.0 correlated to SSAs by V4.2 with R= 0.860, 0.837, 0.855, 0.809 and 0.226 when rm2 = 1.5, 1.8, 2.5 and 3.0 in V5.0 over Qionghai, respectively. The correlation coefficient between SSA by V5.0 and V4.2 was the highest while setting $rm_2$ as 2.0 (the default value) at the two sites.

[Figure]

**Figure 7: Scattergrams of retrieved SSA between SKYRAD V4.2 and V5.0 when $r_{m2}$ =2.0(default), 1.5, 1.8, 2.5 and 3.0 for Qionghai (a) and Yucheng (b) in 2014.$r_{m2}$ represents the model radius for the coarse mode SDF. The green line means the fitted linear regression curve.**

We have added the above figure and comments in the revised manuscript. (P10 Line14- P11 Line 20)

Following the reviewer's suggestion, we have investigated whether the total amount of aerosols in the atmosphere were linked to the difference in SSA between the two versions. As shown in the following figure, the SSA differences between the two versions decreased with the corresponding AODs increased at the two sites.

[Figure]

**Figure 8: Scattergrams of the SSA differences at 500nm between V5.0 and V4.2 (defined as: |SSA_V5.0-SSA_V4.2|) and the corresponding AODs at wavelengths of 500 nm by V5.0 during February 2013 to December 2015.**

We have added the above figure and discussions in the revised manuscript. (P11Line21- P12 Line 11)

We have also investigated the variation of the SSA differences between the two versions with solar height over the two sites. As shown in the following figure, the solar height isn't likely linked to the differences between the two versions.

[Figure]

**Figure: Scattergrams of the SSA differences at 500nm between V5.0 and V4.2 (defined as: |SSA_V5.0-SSA_V4.2|) and the corresponding solar height over Qionghai (a) and Yucheng (b).**

P9 l16-17 Some reference or some data are needed to provide evidence about the meteorological argument.

**Response:** We have added more meteorological description of the two sites and the corresponding references as shown in the above response to the comment ' P3 l19-24 More detailed description of the locations is needed' . (P3Line28- P4 Line 15)

P9 l20 Further discussion and evidence are needed to support this argument.

**Response:** We have added the following discussions in the revised manuscript.

The prevailing winds in Yucheng were from the northwest in winter and spring, and Yucheng was in the downwind of Hebei province where located many industrial enterprises emitted pollutants including secondary inorganic aerosols (Tao et al., 2017; Zhao et al., 2018c).AOD was higher in spring than in autumn and winter likely related to the long-range transportation of dust from northern/northwestern China and pollutants emitted from enterprises in Hebei (Tan et al., 2012; Tao et al., 2017).   (P14Line10- 15)

P10 l14. By definition, SSA will decrease when absorbing aerosols increase. This sentence does not provide any explanation on the behavior. More detailed discussion should be added on these results.

**Response:** We have added the following comments about SSA seasonal variation in the revised manuscript.

The lowest seasonal average SSA was observed in winter, which was probably attributable to the regional transport of the air masses originated from the regions outside of Hainan province in Eastern China, where a great amount of coal was used for industrial enterprises and emitted high concentrations of OC and EC (Liu et al., 2018). In Yucheng, the seasonal pattern of SSA was consistent with AOD, the lowest seasonal average SSAs were also observed in winter due to carbonaceous aerosols increasing by heating activities and biomass burning, seasonal average contributions of carbonaceous aerosols were evidently higher in cold seasons than in warm seasons (Tao et al., 2017). High concentrations of fine particulate nitrate were frequently observed in summer in Yucheng (Wen et al., 2015), likely to cause the high SSA in summer. (P15Line2-10)

Paragraph 3.4.2 Since for SSA the selection of version 4.2 or 5.0 could lead to different conclusions on the type of aerosols, some discussion on that issue should be added here. P10 l26-17 Since the algorithm uses bimodal fits, there was no way to find a different distribution

**Response:** Based on the new experiment results, the SSA and $m_i$ had relatively high correlation coefficients between V4.2 and V5.0 with default $rm_2$ values. In addition, some tests by Hashimoto et al showed that the SDF setting in V5.0 was useful for detecting ill-conditioned data caused by cirrus contaminations, horizontally and/or temporally inhomogeneous aerosol stratification, and so on (Hashimoto et al., 2012). So we still chose the retrieved results by V5.0 to analyze the seasonal variability of the aerosol optical properties over Qionghai and Yucheng. (P12Line12-17)

P11 l2-4. Why anthropogenic aerosols should decrease in winter/spring ? Are there any information on the human activities in the area? Why sea salt aerosols increase? Also, some information about the monsoonal influence in the region should be added for the readers that are unfamiliar with local climatology (preferably at the site description section at page 3)

**Response:** The new seasonal averaged volumes of the different aerosol size distributions (dv/dlnr)

in Qionghai and Yucheng are shown as follows. In Qionghai, the fraction of the fine aerosol particles was much smaller in summer than for the other seasons, the summer meteorological conditions such as high wind speeds, high mixing heights, and the fresh air masses originated from or passed through the sea, which may be contributable to the decrease of pollutant concentrations (Liu et al., 2018) and introducing some sea salt particles of a relatively large size. As shown in the following figure, the seasonal averaged peak of fine mode and coarse mode were both in winter, the air masses of transport were mainly originated from the mainland China, fine and coarse particle were both long range transported to Qionghai in winter (Wu et al., 2011; Liu et al., 2018).

[Figure]

**Figure 11: Seasonally averaged volumes of the different aerosol particle size distributions based on SKYRAD V5.0 over Qionghai (a) and Yucheng (b) for the period from February 2013 to December 2015.**

We have added the above figure and discussion in the revised manuscript. (P15Line18-26)

P11 l6-8 Also fine mode is very high in summer (compared to Qionghai). Any interpretation on that? The only source of large particles in the area is dust long transport or are there any other sources?

**Response:** High concentrations of fine particulate nitrate were frequently observed in summer in Yucheng (Wen et al., 2015), likely to cause the high SSA in summer. The prevailing winds in Yucheng were from the northwest in winter and spring, Yucheng was in the downwind of Hebei province where located many industrial enterprises emitted pollutants including secondary inorganic aerosols (Tao et al., 2017; Zhao et al., 2018c).The aerosol was not only from winter heating but also from regional transport, the fine-mode and coarse-mode particles was both high in

winter in Yucheng. The volume of the coarse aerosol particles relative to the whole was much larger than for the other seasons in spring in Yucheng probably because of the presence of the dust particles transported from the northwest of China and secondary inorganic aerosols emitted from enterprises in Hebei (Tao et al., 2017).

We have added the above discussion in the revised manuscript. (P14 Line10-15; P15 Line28-P16Line2)

P11 l8. Also fine mode seemed to peak to almost double values in summer/winter compared to spring/autumn. Is there are any explanation for this behavior?

**Response:** Yucheng and the surrounding areas are famous for their agriculture (e.g., wheat and corn) and grazing land (e.g., donkeys and chickens). In addition, the site near 20 to 30 km radius located several factories in the production of inorganic and organic fertilizers (Wen et al., 2015), and the application of fertilisers to farmland emitted a great deal of NH3 (Zhao et al., 2012). High concentrations of fine particulate nitrate were frequently observed in summer in Yucheng (Wen et al., 2015), likely to cause the high SSA in summer. (P4 Line9-13; P15 Line9-10)

Figure 6. X axes label is missing

**Response:** In all the new figures, we have checked X axes label and other details.

General comment for 3.4 I suggest to summarize the types and variations of aerosols in both sites in a more descriptive way at the end. Also, It would be very useful a discussion –based on earlier paragraphs- on the properties and conditions that both version come together and the conclusions that have higher uncertainties due to the deviations between the algorithms.

**Response:** Based on the new experiment results, the AOD, SSA and $m_i$ had relatively high correlation coefficients between V4.2 and V5.0. Quality control and cloud screening procedures in V5.0 are stricter than V4.2 (Hashimoto et al., 2012). So we still chose the retrieved results by V5.0 to analyze the seasonal variability of the aerosol optical properties over Qionghai and Yucheng. (P12 Line12-17)

P12 l16-18 There is no evidence in the study of the cause of this behavior (algorithm or type of

aerosols?). More work should be done before coming to this conclusion.

**Response:** We have replaced this conclusion with the following conclusion based on the above analysis. The location and distribution of major industrial sources, intensity of local minor sources such as winter heating, and prevailing wind directions together caused the slightly different magnitudes of seasonal variations among the two sites discussed above. (P18 Line10-12)

Reviewer #2

This analysis is divided in two parts: first, a comparison of aerosol properties retrieved by inversion code SKYRAD versions 4.2 and 5.0 is performed, based on two years of data for two SKYNET sites. Second, version 5.0 is used to analyze the aerosol characteristics at the two sites. This kind of study is needed for the improvement of the SKYNET network methodology, and also for the improvement of our knowledge of the aerosol characteristics at China. Therefore, it is adequate for this journal.

However, I would recommend to accept the paper after a major revision, mainly related to: adding detail to the text - improving the graphical representations - further discussing the temporal behavior of aerosol at the two sites

The use of English is adequate, although some flaws are pointed out and I would recommend a revision.

**Response:** Thank you for your valuable comments and constructive suggestions. We have carried out additional experiments and found that the calibration constants in the previous experiments were incorrect, so we corrected them and re-carried out the experiments and numerical tests; some results and figures have been updated and represented in the following response.

We have added more details related to the graphical, climate and major chemical compositions in PM2.5 in the two sites. Some of the new figures and comments about inter-comparisons results between V5.0 and V4.2 have been shown in the following comments. Meanwhile, we have also investigated some parameters linked to the SSA differences between the V5.0 and V4.2, the seasonal variation of aerosol have been discussed combining the possible emission sources and prevailing wind based on more data and references as shown in the following comments.

**General comments:**

Introduction: Some more background discussion would be welcome. Please add also a few comments about general differences between versions 4.2 and 5.0. Non-sphericity, minimization technique used, etc, so general readers can learn about these codes.

**Response:** Non-sphericity particle model are neither included in V4.2 nor V5.0. We have added the comment in the revised manuscript. (P5 Line 8)

V4.2 uses the iterative relaxation method of Nakajima et al. (1983, 1996) to remove the multiple scattering contributions and derive an optimal solution using a statistical regularization method (Turchin and Nozik, 1969) by minimizing the cost function as proposed by Phillips (1962) and Twomey (1963).

V5.0 uses the non-linear maximum likelihood method defined by Rodgers (2000) which was based on the Bayesian theory. The non-linear inversion has a strong dependence on the estimation of the first-guess solution. Version 5.0 uses an a priori SDF of a bimodal log-normal function.

Section 2.1: Please explain the method used for the calibration, and any findings you consider interesting to note, if any (calibration drift, etc).

**Response:** The Improved Langley (IL) plot method is used in this study to determine the temporal and spectral calibration constants for direct intensity (F0) with accuracy of about 1.0–2.5 %, depending on the wavelength (Nakajima et al., 1996; Campanelli et al., 2004). The calibration by IL plot method is made daily, the variation of F0 due to instrumental drift can be quickly spotted, and then appropriate corrections to data can be applied exactly from the period in which the deviation occurred (Campanelli et al., 2004). We have added the above comments in the revised manuscript. (P3 Line20-25)

It is important also to detail the description of the two sites, including a map if possible, to understand aerosol characteristics. -

**Response:** We have added a map in the revised manuscript to show the locations of the two SKYNET sites in this study. (P4 Line18)

In section 2.2 Site description, we have added some more details including monsoon, temperature, and precipitation. (P3 Line 26- P4 Line 16)

In addition, based on the results simulated by the Community Multi-scale Air Quality model with

the 2D Volatility Basis Set (CMAQ/2D-VBS) (23), we have added section 3.2.1 to describe the major chemical compositions in PM2.5 and their percentage contribution to PM2.5 at the two sites. (P13 Line3-17)

Section 2.2: Please cite the source for the details given about version 5.0. Comments about expected errors would be useful at this stage.

**Response:** The non-linear maximum likelihood method used in V5.0 has a strong dependence on the estimation of the first-guess solution. Version 5.0 uses an a priori SDF of a bimodal log-normal function. The reference 'Hashimoto et al., 2012' gives more details about V5.0. They had performed various test simulations with SKYRAD.pack V4.2 and V5.0 (Hashimoto et al., 2012), and found: In the case of a large amount of coarse particles with radius greater than 10 μm existing , the numerical tests performed by Hashimoto et al showed that V4.2 could retrieve the SDF relatively well, including the coarse mode, in comparison with V5.0, because the smoothness condition given by Eq. (2) allowed the retrieved SDF to be distributed beyond 10 μm radius, on the other hand, V5.0 underestimated the coarse mode of the SDF because of the strong SDF constraint condition given by Eq. (5) with a small model radius $rm_2 = 2.0$ μm for the coarse mode SDF (Hashimoto et al., 2012). So we have compared the differences between retrieved SSAs at 500 nm byV5.0 and V4.2 when set rm2 = 1.5, 1.8, 2.0(default), 2.5 and 3.0 in Skyrad.pack V5.0 based on the measurements in 2014. As shown in Fig.7, SSAs by V5.0 correlated to SSAs by V4.2 with R= 0.860, 0.837, 0.855, 0.809 and 0.226 when $r_{m2}$ = 1.5, 1.8, 2.5 and 3.0 in V5.0 over Qionghai, respectively. The correlation coefficient between SSA by V5.0 and V4.2 was the highest while setting $r_{m2}$ as 2.0 (the default value) at the two sites.

[Figure]

**Figure 7: Scattergrams of retrieved SSA between SKYRAD V4.2 and V5.0 when $r_{m2}$ =2.0(default), 1.5, 1.8, 2.5 and 3.0 for Qionghai (a) and Yucheng (b) in 2014.$r_{m2}$ represents the model radius for the coarse mode SDF.**

We also investigated whether the total amount of aerosols in the atmosphere were linked to the difference in SSA between the two versions. As shown in Fig. 8, the SSA differences at 500nm between the two versions (defined as: |SSA_V5.0$_{500nm}$ - SSA_V4.2$_{500nm}$ |) decreased while the corresponding AODs at wavelengths of 500 nm by V5.0 increased at the two sites. When the AOD was high (in this study the threshold was set to 0.5 for AOD$_{500nm}$), SSA retrieved by V5.0 had a good comparison with those with V4.2. It is well known that the inversion products have a very high uncertainty in cases of very low aerosol burdens, the retrieval error in SSA rapidly increases with decreasing AOD (Dubovik et al., 2000), especially in parameters such as the imaginary part of the refractive index.

We have added the above comments and figure in the revised manuscript. (P10 Line14- P12 Line11)

Section 3.1: It is possible to further analyze the comparison of the SDF, including some statistics. In the first part of the paper, perhaps the authors should focus on the analysis of the differences (absolute or relative) and leave the absolute retrievals for the second part of the study (analysis of the aerosol properties). Why AOD is not included in the comparison?

**Response:** Following the reviewer's suggestion, we have added the following figure which showed the plots of AOD values at each wavelength derived from the solar direct irradiance between

the two versions. High correlation was found with a significant coefficient larger than 0.995 at each band except 1020nm over Qionghai. High consistency of AODs between V4.2 and V5.0 indicates that the inversion process in V5.0 did not bring about a large change in the retrieved direct solar radiation (Hashimoto et al., 2012).

[Figure]

Figure 2: Scatter plot and correspondent linear fitting for the aerosol optical depth (AOD) between SKYRAD V4.2 and V5.0 at wavelengths of 400, 500, 670, 870, and 1020 nm over Qionghai and Yucheng during February 2013 to December 2015.

We have added the above comments and figure in the revised manuscript. (P6 Line32- P7 Line7)

We have replaced Fig.1 with the following figure which shows the retrieved monthly volume size distribution between SKYRAD V4.2 (dotted lines) and V5.0 (solid lines) for Qionghai (blue line) and Yucheng (red lines) during February 2013 to December 2015.

[Figure]

Figure 3: Retrieved monthly volume size distribution between SKYRAD V4.2 (red lines) and V5.0 (blue lines) for Qionghai (dotted lines) and Yucheng (solid lines) during February 2013 to December 2015

We have added the above figure and comments in the revised manuscript. (P7 Line12- P8Line4)

As shown in the following figure, the differences between the volumes retrieved by V4.2 and by V5.0 at each bin (d ($V_{V4.2}$- $V_{V5.0}$) /dlnr) were averaged monthly over Qionghai and Yucheng, the differences of the volume size distributions between the two versions were obviously very large for the coarse mode with a radius of over 5 μm in most months.

[Figure]

**Figure : The difference between retrieved monthly volume size distribution by V4.2 and by V5.0 ( d($V_{v4.2}$- $V_{v5.0}$) /dlnr ) for Qionghai (blued lines) and Yucheng (red lines) during February 2013 to December 2015**

Section 3.2. and 3.3.: similarly to 3.1, concentrate on differences rather than absolute values. Finally, add your opinion about the most adequate version to use in the remaining, based on the results, so both parts of paper are smoothly linked.

**Response:** Based on the new experiment results as shown in the following figures, the SSA and $m_i$ had relatively high correlation coefficients between V4.2 and V5.0 with default $r_{m2}$ value based on the above comparison results. In addition, some tests by Hashimoto et al showed that the SDF setting in V5.0 was useful for detecting ill-conditioned data caused by cirrus contaminations, horizontally and/or temporally inhomogeneous aerosol stratification, and so on (Hashimoto et al., 2012). So we still chose the retrieved results by V5.0 to analyze the seasonal variability of the aerosol optical properties over Qionghai and Yucheng. (P12 Line12- 17)

[Figure]

**Figure 4: Scattergrams of the single scattering albedo between SKYRAD 4.2 and 5.0 at wavelengths of 400, 500, 670, 870, and 1020 nm over Qionghai and Yucheng during February 2013 to December 2015. Only data with $AOD_{500nm}>0.2$ are shown. The green line means the fitted linear regression curve.**

[Figure]

**Figure 5: Scattergrams of the imaginary part of the complex refractive index ($m_i$) results between SKYRAD 4.2 and 5.0 at wavelengths of 400, 500, 670, 870, and 1020 nm over Qionghai and Yucheng during February 2013 to December 2015.**

Section 3.4: I think the analysis of the aerosol properties at the two sites need a deeper analysis, also including references to previous analysis from China or elsewhere.

**Response:** Based on the new experiment results, we have made major revision on this section. Section 3.1, 3.2 , 3.3 have been merged into Section 3.1, Section 3.4.1, 3.4.2, 3.4.3, 3.4.4 have been changed into Section 3.2.2, 3.2.3, 3.2.4 , 3.2.5 as follows.

First, we have added section 3.2.1to describe the major chemical compositions in PM2.5 at the two sites. (P13 Line3- 17)

We have analyzed the seasonal variability of the aerosol optical properties at the two sites at a deeper level based on more references in section 3.2.2, 3.2.3, 3.2.4 and 3.2.5. (P13 Line23- P16 Line20)

Line 251 is particularly vague, as other reasons for the increase of AOD in summer are usually considered (differences in transport from remote areas, increase of secondary aerosols due to higher solar radiation...). In contrast to first part of the paper, in the second part I would recommend to focus on the absolute values, represented in monthly means along the year, with corresponding boxplots, for example. Current analysis based on seasonal averages alone, is not optimum.

**Response:**

The increase of AOD in summer in Yucheng maybe was caused by hygroscopic effects. Yucheng and the surrounding areas are famous for their agriculture and grazing land. In addition, the site near 20 to 30 km radius located several factories in the production of inorganic and organic fertilizers (Wen et al., 2015), and the application of fertilisers to farmland emitted a great deal of NH3 in summer (Zhao et al., 2012). The humidity of Yucheng (belong to Shandong province) is highest in summer than other seasons (Meng et al., 2007). High humidity combined with large fractions of hygroscopic chemical components can enhance light extinction and haze intensity the scattering coefficient of secondary inorganic aerosols (such as sulfate, nitrate and ammonium) (Tao et al., 2017). (P4 Line9-13; P14 Line 6-10)

We have tried to represent the monthly means with corresponding boxplots as follows. The references to previous analysis from China or elsewhere related to the two sites are mostly based on seasonal averages. There are no other data available to be inter-compared with our results during the experiment time. To analyze the reasons for the variation or inter-compare based on the references, the temporal variation analysis are based on seasonal averages as above.

[Figure]

**Figure : Monthly variations in the AOD based on SKYRAD V5.0 over Qionghai (a) and Yucheng (b) for the period from February 2013 to December 2015. The boxes represent the 25th to 75th percentiles of the distributions while the dots and solid lines within each box represent the means and medians, respectively.**

[Figure]

**Figure : Monthly variations in the single scattering albedo (SSA) based on SKYRAD V5.0 over Qionghai (a) and Yucheng (b) for the period from February 2013 to December 2015. The boxes represent the 25th to 75th percentiles of the distributions while the dots and solid lines within each box represent the means and medians, respectively.**

[Figure]

**Figure : Monthly variations in the imaginary part of the refractive index based on SKYRAD V5.0 over Qionghai (a) and Yucheng (b) for the period from February 2013 to December 2015. The boxes represent the 25th to 75th percentiles of the distributions while the dots and solid lines within each box represent the means and medians, respectively.**

**Other specific corrections:**

line 59: many -> several? -

**Response:** We have replaced "many" with "several".    (P2 Line14)

line 74: There are a few -

**Response:** We have added "a" before "few".    (P2 Line 29)

line94: The dynamic range seems should be 10^7 instead of 107? -

**Response:** We have replaced "107" with "$10^7$".    (P3 Line 19)

line 120-121: rewrite (parenthesis?) -

**Response:** We missed a right parenthesis. We have added it in the revised manuscript as below.

(i.e., $\ln(r_{i+1}) - \ln(r_i) = const$)    (P5 Line 19)

line 131-132: e^2 -

**Response:** We have replaced "e2" with "$e^2$".    (P5 Line 30)

line 156: more comments on the cloud screening and quality control -

**Response:** SKYRAD V5.0 developed a stricter data quality control method of observation data and cloud screening. The standard process of quality control in SKYNET applies a retrieval error between observations and calculated theoretical values by using retrieval values, $\sigma_{obs}$

$$\sigma_{obs} = \sqrt{W_e \sum_i (\frac{\tau_{\lambda_i}}{\tau_{\lambda_i}^{meas}} - 1)^2 + W_p \sum_i \sum_j [R_{\lambda_i}(\Theta_j) / R_{\lambda_i}^{meas}(\Theta_j) - 1]^2}$$

where ($\tau_{\lambda_i}^{meas}$ and $R_{\lambda_i}^{meas}$) and ($\tau_{\lambda_i}$ and $R_{\lambda_i}$) are measured and retrieved observation vectors for the AOD and relative sky radiance, $N_i$, $N_j$, and $N_{total} = N_i + N_i \times N_j$ indicate the number of measured wavelengths, scattering angles, and their total, respectively, $W_e = W_P = 1/N_{total}$. In V4.2, the data if the value of $\sigma_{obs}$ is larger than 0.2, but $\sigma_{obs}$ is set 0.07 as a threshold for data rejection in V5.0. There are some other differences between V4.2 and V5.0 on the issue of quality control of observation data and cloud screening (Hashimoto et al., 2012).

We have added the above comments in the revised manuscript. (P6 Line 16-25)

line 173: it is important to highlight the fact that the unrealistic coarse mode in v4.2 is removed -

**Response:** The unrealistic coarse mode in v4.2 is removed in V5.0 by the constraint of a reduced SDF for particles with radius greater than 10 μm. Some tests by Hashimoto et al showed that the SDF setting in V5.0 was useful for detecting ill-conditioned data caused by cirrus contaminations, horizontally and/or temporally inhomogeneous aerosol stratification, and so on (Hashimoto et al., 2012).

We have added the above comments in the revised manuscript. (P12 Line 12-14)

line 244: The AOD is -

**Response:** We have replaced "was" with "is". (P13 Line 19)

line 289-291: three significant digits is enough for the refractive index (1.45 etc)

**Response:** Following the reviewer's suggestion, the numbers referred to the real part of the refractive index in the revise manuscript have been changed to be with three significant digits. (P16 Line 11-13)

[revised manuscript text omitted]

---

## Author Response (AR2)

We appreciate the reviewers' valuable comments and constructive suggestions which help us improve the quality of the manuscript. We have carefully revised the manuscript according to these comments. Point-to-point responses are provided below. The reviewers' comments are in black, our responses are in blue and changes in manuscript are in red.

Reviewer:

Authors have a made a huge effort by rerunning the algorithms in order to improve the study, also a lot of details about the two measuring sites and their atmospheric conditions has been added. However, my major concerns have not been answered and although a lot of information has been added, I can quote from my earlier review:

"However, the manuscript lacks of explanations on the causes of differences between the v4.2 and v5.0 retrievals and sufficient evidence on the actual seasonal variability of aerosols in the two regions. Algorithms of both versions are clearly described, but it is crucial to pinpoint and discuss the way the differences between the versions affect the retrievals. Since the two algorithms are not treated as a "black box", it should be more clear which physical processes affects the retrievals and which atmospheric conditions could lead to highest uncertainty. At least some discussion on the uncertainty of each variable in each approach should be provided. Also, the part about the seasonal variability of aerosols properties, results are presented but not investigated and discussed in the level expected for a scientific study."

I think nothing has been done in this direction. Also, information added are not used in order to interpret the results and explain the findings of the present study. Also, some discussion is needed to conclude the validity or higher quality of v5.0 retrievals before using them for aerosol characterisation in section 3.2

I suggest a major revision of the manuscript before considering for publication in AMT.

**Response:** Thank you for your valuable comments and constructive suggestions.

V4.2 is based on inversion scheme of the Phillips-Twomey type solution of the first kind of Fredholm integral equation with homogeneous smoothing constraint, and V5.0 is based on the second kind of the equation with inhomogeneous constraint with a priori information for aerosols (Twomey, 1963) to retrieve the inherent aerosol optical properties.

The most different physical process between V4.2 and V5 is a derivation of particle size

distribution. V4.2 doesn't have a constraint for the size distribution. On the other hand, V5.0 has a constraint for it using the term $(S_a^{-1}(x_k - x_a))$ in Eq. 4 and gives a strong constraint for the edge of size distribution, so the edge of size distribution in V5.0 close to zero, but V4.2s' looks jumped.

We have compared the differences between retrieved SSAs at 500 nm byV5.0 and V4.2 when set the coarse model radius rm2 in Eq. (5) as rm2 = 1.5, 1.8, 2.0(default), 2.5 and 3.0 in Skyrad.pack V5.0 based on the measurements in 2014. Based on the sensitivity tests, we found the correlation coefficient between SSAs by V5.0 and V4.2 was the highest when setting rm2 as 2.0 (the default value) in V5.0 at the two sites.

We assumed an error of ±5% for calibration constant F0, ±5% for solid view angle SVA, ±50% (±0.05) for ground surface albedo Ag. We compared the differences in retrieved SSA values at a wavelength of 0.5 μm between cases with and without the assumed errors. On the basis of the sensitivity tests, it is concluded that an error in the calibration constant (F0) causes an error in both retrieved SSA and AOD. The averaged differences in retrieved SSA values due to ±5% error in F0 varied from 3% to 5%. An error of ±5% for solid view angle SVA introduced about ±2% differences in retrieved SSA values both by V4.2 and V5.0. Overestimation or underestimation in the Ag results in underestimation or overestimation of the SSA. An error of ±50% for ground surface albedo Ag caused about 1% averaged differences in retrieved SSA values both by V4.2 and V5.0. With the atmospheric pressure PRS increased by 1%, 2%, 3% and 4%, the averaged differences in SSAs didn't exceed 0.8%.

We also investigated whether the total amount of aerosols in the atmosphere were linked to the differences in SSA between the two versions. It could be said that the condition of low AOT affected the retrieval accuracy of SSA, especially when AOT is less than 0.4.

**Specific comments**

P3l25 Is the same calibration used also for sky radiance measurements?

**Response:** The calibration method for sky radiance measurements is different from the calibration method for the direct solar irradiance measurements. The solar disk scan method has been routinely used in the SKYNET measurement of the SVA of the sky radiometer by scanning a circumsolar domain (CSD) of ±1° around the sun with every 0.1° interval (Nakajima et al., 1996; Uchiyama et al., 2018). (P4 Line 1- 5)

P6 l23-25 Other differences should at least be mentioned, before referring to Hashimoto et al work.

At the end the present manuscript is about these differences.

**Response:** Other differences between V5.0 and V4.2 are mainly the cloud screening algorithm. The cloud screening method in V4.2 relies heavily on the global flux test and needs global irradiance data, V5.0 poses a condition regarding the magnitude of the coarse mode of the SDF as follows:

$$C_v \times v(2.4\,\mu m) < \max\{v(7.7\,\mu m), v(11.3\,\mu m), v(16.5\,\mu m)\}.$$

where Cv is a threshold coefficient to be determined for optimum rejection of cirrus contamination, v(r) is vertically integrated aerosol SDF, as a function of particle radius, r. Based on the analysis of data at the Pune and Beijing sites (Hashimoto et al., 2012), Cv is set as 2 in V5.0 to reject most cirrus contamination cases and pass through dust cases. It is necessary to determine Cv after collecting more cirrus contamination data and dust day data. (P7 Line 5- 15)

**Section 3.**

AOD is derived from Direct Irradiance measurements. In earlier text only the algorithms for inversion products is described. Are there differences in direct sun algorithms between 5.0 and 4.2? Since AOD is presented these details should be explained. Also, even this slight differences of AOD could propagate considerable uncertainties, thus it should be more clear if and how AOD is used in inversion calculations in both versions.

**Response:** There are nearly no differences in direct sun algorithms between 5.0 and 4.2. As shown in Fig.2, there were very slight differences between AODs by V5.0 and V4.2, this is mainly caused by the very small differences in calibration constant F0. F0 in V4.2 and V5.0 are both determined from sky radiance data by the Improved Langley method. V5.0 adopts more rigorous data processing and cloud detection methods. The sky radiance measurements which involved in F0 calculation are a little different in V4.2 and V5.0. (P7 Line 27- 30)

In inversion calculations in V4.2 and V5.0, AOD are both used as indicative values in the first step of the loop but are updated at each iteration. The AOD data can be given different weights with respect to the normalized diffuse sky flux data, according to their reliability. (P8 Line 1- 4)

**Section 3.1.1**

As it is stated now, it seems that the differences are caused by just the different approaches in the versions and the functions that the data are fitted to. Is that the conclusion? Are these differences

connected to atmospheric conditions? Additionally, these approach of plotting just mean monthly values hides the real picture. At least some basic statistics (average and standard deviation of the differences at each bin) should be presented and discussed. Currently there is no information on the scattering of differences and if these differences are permanent biases.

**Response:**

Based on the study, we found that the most different physical process between V4.2 and V5 is that V5.0 introduced a constraint for the size distribution for it using the term (Sa-1(xk – xa)) in Eq. 4 and gives a strong constraint for the edge of size distribution, the values of the retrieved size distribution of the smallest size classes (r<0.05 μm) and the largest size classes (r > 10 μm) by V5.0 were close to zero.

We have added the scattering of VOL differences in the manuscript as shown in Fig.4 and Fig.5. We have also added some basic statistics (average and standard deviation of the differences at each bin) in Table 1 and Table 2. The percentage difference of the volume size distribution between SKYRAD V5.0 and V4.2 were larger than 50% at smaller size (r <0.025 μm at Qionghai, r <0.017 μm at Yucheng) and larger size (r >10μm at both sites). When the radius is between 0.17-5 μ m, the size distributions retrieved by V5.0 were in good agreement with those by V4.2. (P9 Line 5- 14)

**Section 3.1.2**

At this section there is no discussion on the causes of the differences. g. Are they explained strictly algorithmically or is there some natural process driving them? In most cases v5.0 retrieves lower SSA values suggesting the presence of more absorbing aerosols. Is there any evidence on that?

**Response:** In most cases V5.0 retrieving lower SSA values couldn't suggest the presence of more absorbing aerosols.V5.0 tends to underestimate the SSA due to underestimation of the coarse aerosols when the a priori SDF for constraint tends to be close to zero for radii larger than 10μm.

[Figure]

**Fig. 8.** The percentage difference of the relative radiances at 0.5 µm for each scattering angle between SDFs with and without particles over 10 µm in radius.

The above figure from the reference Hashimoto et al. (2012) shows that the difference between the relative intensity with and without a cut above 10 µm for the SDF (R is the relative radiance, ∆R=[R(cut above 10µm)- R(no cut above 10µm)]/ R(no cut above 10µm)). From this result, the lack of a large coarse part in the SDF causes overestimation of sky radiance at all observation angles. It is likely that V5.0 works to decrease the SSA value to dim the sky radiance in the calculation when a tight constraint on the SDF for particles with radius over 10 µm is applied. (P11 Line 17- 23)

P8l12-16 Why at Qionghai are only relative differences presented, while for Yucheng both relative and absolute are mentioned? Absolute differences are more important generally and it should be mentioned for both datasets. Also, as mentioned above, at least a standard deviation of these differences should be presented. From the scatter plots it is clear that the deviations are very diverse in each dataset.

**Response:** We have updated the tables, as the former table was got based on all the measurements, the new tables are based on the simultaneous observations of V4.2 and V5.0, which is consistent with the scatter plots. The standard deviations of absolute differences were added. (P32 and P33)

Figure 4. At Qionghai at 1020nm, it seems there two groups of measurements. One with values close to 1 and one with values lower than 0.8. Is there a physical explanation for this? It clearly needs more investigation this behavior. The same behavior at 1020 nm is also presented at figure 5 for refractive index.

**Response:** We haven't found a clear reason for this behavior. We need further investigation in the

future work.

**Section 3.1.3**

Same as above. At least some discussion on the scattering of the data, since real part appears to have almost random differences. . Are the differences explained strictly algorithmically or is there some natural process driving them?

**Response:** In the retrieval, V4.2 found the optimum complex refractive index by trying several refractive indices. Complex refractive index in V4.2 can only be chosen from the predefined set of values in V4.2. In V5.0, complex refractive index were directly included in the state vector x, including constraints on the complex refractive index. As a priori estimation, mr usually be set as 1.5. (P12 Line 18- 21)

   At present, we also haven't found a clear reason for the fact that real part appears to have almost random differences. We need further investigation in the future work.

**Section 3.1.4**

This sensitive test is an appropriate way to understand the algorithmical differences. But still no conclusion is drawn from this test and nothing is discussed in respect to the findings of previous sections.

**Response:** We have added some sensitivity tests for the main causes of error in the SSA and AOD retrieval by V5.0 and V4.2.We assumed an error of ±5% for calibration constant F0, ±5% for solid view angle SVA, ±50% (±0.05) for ground surface albedo Ag. We compared the differences in retrieved SSA values at a wavelength of 0.5 μm between cases with and without the assumed errors. On the basis of the sensitivity tests, it is concluded that an error in the calibration constant (F0) causes an error in both retrieved SSA and AOD. The averaged differences in retrieved SSA values due to ±5% error in F0 varied from 3% to 5%. An error of ±5% for solid view angle SVA introduced about ±2% differences in retrieved SSA values both by V4.2 and V5.0. Overestimation or underestimation in the Ag results in underestimation or overestimation of the SSA. An error of ±50% for ground surface albedo Ag caused about 1% averaged differences in retrieved SSA values both by V4.2 and V5.0. With the atmospheric pressure PRS increased by 1%, 2%, 3% and 4%, the averaged differences in SSAs didn't exceed 0.8%. (P14 Line 6- P16 Line 12)

   Base on the inter-comparison results in Section 3.1 and the sensitivity tests in Section3.2, we

couldn't get the conclusion that V5.0 is definitely better than V4.2. We haven't yet got other measurements in the two sites to help us prove that V5.0 is better than V4.2. The most different physical process between V4.2 and V5 is a derivation of particle size distribution. On the one hand, V5.0 tends to be robust to the cloud contamination, owing to inversion constraint by a priori SDF which filters out coarse particles to simulate cloud-scattered radiation. Some tests by Hashimoto et al (2012) showed that the SDF setting in V5.0 was useful for detecting ill-conditioned data caused by cirrus contaminations, horizontally and/or temporally inhomogeneous aerosol stratification, and so on (Hashimoto et al., 2012). On the other, due to a priori SDF for constraint tends to be zero for radii larger than 10μm, V5.0 will underestimate the coarse mode aerosols when a large amount of coarse particles of the dust-like aerosol type with radius greater than 10 μm exits. Estellés et al. (2018) found underestimation of the coarse aerosols by the V5.0 in African dust storm cases, whereas V4.2 retrieved coarse mode SDF similar to the observed one (Estellés et al., 2018). (P18 Line 4- 16)

Considering that V5.0 adopts more rigorous data processing and cloud detection methods, and the SSA and mi had high correlation coefficients between V4.2 and V5.0 with default the coarse mode radius rm2 value in V5.0 based on the above comparison results, we chose the retrieved results by V5.0 to analyze the seasonal variability of the aerosol optical properties over Qionghai and Yucheng. (P18 Line 17- 21)

We have added the above comments in the revised manuscript.

Figure 8. This approach also helps to increase the understanding of the algorithm. I suggest to plot real differences instead of absolute, because the sign is important to understand weather the version over or underestimates compared to the previous one. Also the lack of discussion on the uncertainty of the retrievals in the manuscript, makes it harder to interpret which range of difference is in the expected uncertainty.

**Response:** Following the reviewer's suggestion, we have replaced the absolute with real differences. (P17 Figure 10)

P13 l1-2. Also, why selecting v5.0 for studying the seasonal variability should be explained here.

**Response:**

Base on the inter-comparison results and the sensitivity tests, we couldn't get the conclusion

that V5.0 is definitely better than V4.2. We haven't got other measurements in the two sites to help us prove that V5.0 is better than V4.2. On the one hand, V5.0 tends to be robust to the cloud contamination, owing to inversion constraint by a priori SDF which filters out coarse particles with radius greater than 10 μm. Some tests by Hashimoto et al (2012) showed that the SDF setting in V5.0 was useful for detecting ill-conditioned data caused by cirrus contaminations, horizontally and/or temporally inhomogeneous aerosol stratification, and so on (Hashimoto et al., 2012). On the other, due to a priori SDF for constraint tends to be zero for radii larger than 10μm, V5.0 will underestimate the coarse mode aerosols when a large amount of coarse particles of the dust-like aerosol type with radius greater than 10 μm exits. Estellés et al. (2018) found underestimation of the coarse aerosols by the V5.0 in African dust storm cases, whereas the version 4.2 retrieved coarse mode SDF similar to the observed one. (P18 Line 4- 16)

Considering that V5.0 adopts more rigorous data processing and cloud detection methods, and the SSA and mi had high correlation coefficients between V4.2 and V5.0 with default the coarse mode radius rm2 value in V5.0 based on the above comparison results, we chose the retrieved results by V5.0 to analyze the seasonal variability of the aerosol optical properties at the two sites. (P18 Line 17- 21)

**Section 3.2.1**

I honestly have a difficulty understanding why these section for PM2.5 have been added Surely it makes more clear the local emissions types, but it seems unlinked with the rest of the study. Findings mentioned here are nowhere used to explain anything about SKYNET retrievals and their behavior. Also, it is an unexplained decision to study PM 2.5 while from all SKYNET retrievals there is a picture of constant dominance of larger particles in both regions. Unless you could integrate the findings to the discussion in the rest of the study, linked differences found between the two sites and preferably even connect the deviations between the two versions with the types of aerosols, I suggest removing this section.

**Response:** Thank you for your kind comments. We have removed this section.

**Section 3.2.2**

Which months are considered in each season should be defined. Also, the discussion about the

humidity is not clear. It seems that aerosol loads are generally in the same order throughout the year and humidity causes the variations of AOD. This needs more evidence to support it and a lot of discussion and data are needed to prove it. If this is not the case, please restate to make clear the finding of this analysis.

**Response:** Four seasons were considered in this paper (i.e., spring (March-May), summer (June-August), autumn (September-November), and winter (December-February)) to investigate the seasonal variations of the aerosol optical properties over Qionghai and Yucheng. (P18 Line 24-26)

We have restated the analysis as follows:

The maximum AOD average of 0.99 occurring in summer, several factories which produced inorganic and organic fertilizers located, the stronger sunlight in summer accelerated the photochemical reaction and enhanced the formation of fine particulate nitrate (Wen et al., 2015), the humidity in summer is higher than other seasons over Yucheng (Meng et al., 2007), high humidity combined with large fractions of hygroscopic chemical components (e.g. sulfate, nitrate, ammonium, and some organic matters) can enhance light extinction and haze intensity the scattering coefficient of secondary inorganic aerosols (such as sulfate, nitrate and ammonium) (Tao et al., 2017) .(P19 Line10-16)

**Section 3.2.3**

The discussion in this paragraph is not consistent with the next section. Since lower SSA values in winter are explained by the presence of carbonaceous particles, why winter SDF are dominated by coarse mode in both regions?

**Response:** The volume size distribution in both sites presented bimodal patterns with a 0.1-0.2 μm fine particle mode and a 3.0-6.0 μm coarse particle mode in four seasons, and the volume of the coarse aerosol particles relative to the whole was larger, especially in Yucheng. Carbonaceous particles have higher values than other seasons in both sites due to winter heating and regional transport, black carbon aerosols are dominated by absorption effect, so the seasonal SSAs in winter are lower than other seasons in winter, but it hadn't changed the fact that the two regions are dominated by coarse particles. With winter heating, due to incomplete combustion, in addition to black carbon aerosols, there will also be some dust.